# Homeotic compartment curvature and tension control spatiotemporal folding dynamics

Aurélien Villedieu[1,2], Lale Alpar[1,2], Isabelle Gaugué[1,2], Amina Joudat[1,2], François Graner[3], Floris Bosveld [1,2,4] ✉ & Yohanns Bellaïche [1,2,4] ✉

Shape is a conspicuous and fundamental property of biological systems entailing the function of organs and tissues. While much emphasis has been put on how tissue tension and mechanical properties drive shape changes, whether and how a given tissue geometry influences subsequent morphogenesis remains poorly characterized. Here, we explored how curvature, a key descriptor of tissue geometry, impinges on the dynamics of epithelial tissue invagination. We found that the morphogenesis of the fold separating the adult *Drosophila* head and thorax segments is driven by the invagination of the Deformed (Dfd) homeotic compartment. Dfd controls invagination by modulating actomyosin organization and in-plane epithelial tension via the Tollo and Dystroglycan receptors. By experimentally introducing curvature heterogeneity within the homeotic compartment, we established that a curved tissue geometry converts the Dfd-dependent in-plane tension into an inward force driving folding. Accordingly, the interplay between in-plane tension and tissue curvature quantitatively explains the spatiotemporal folding dynamics. Collectively, our work highlights how genetic patterning and tissue geometry provide a simple design principle driving folding morphogenesis during development.

During development, folding transforms epithelial cell sheets into complex three-dimensional (3D) structures. It underlies morphogenetic processes such as gastrulation, neurulation, and organogenesis[1,2]. Furthermore, folding can morphologically delineate tissue compartments, and fold positions often correlate with a selector or homeotic gene patterns as observed in vertebrate rhombomeres and in insect appendages or body segments[3–6]. As with any morphogenetic process, folding could result from the complex interplay between cell and tissue mechanical stress, mechanical properties, and organ geometry. So far, numerous studies have explored how mechanical force production and tissue mechanical properties control the dynamics of tissue invaginations[7,8]. These studies have defined several mechanisms associated with fold formation including (i) cell apical constriction for

which the surface area or anisotropy of the constricting domain could modulate fold shape;[9–18] (ii) apical-basal shortening of the cells;[19–22] (iii) cell delamination;[23,24] and (iv) tissue buckling due to compressive stresses[11,25–31]. Despite these advances, whether and how epithelial folding dynamics are regulated by curvature, an intrinsic property of any epithelia, has remained unexplored during development.

In this work, by studying *Drosophila* adult cervix (neck) morphogenesis, we initially aimed to understand how the insect body plan is morphologically compartmentalized by deep invaginations. These tissue invaginations are much larger than the cell height and their positions may correlate with homeotic gene expression patterns[6]. We, therefore, envisioned that studying the inter-segmental invagination at the head–thorax region could provide insights into how large

[1]Institut Curie, PSL Research University, CNRS UMR 3215, INSERM U934, F-75248 Paris Cedex 05 Paris, France. [2]Sorbonne Universités, UPMC Univ Paris 06, CNRS, CNRS UMR 3215, INSERM U934, F-75005 Paris, France. [3]Université Paris Cité, CNRS, Matière et Systèmes Complexes, F-75006 Paris, France. [4]These authors contributed equally: Floris Bosveld, Yohanns Bellaïche. ✉e-mail: floris.bosveld@curie.fr; yohanns.bellaiche@curie.fr

out-of-plane displacements are generated, and whether and how homeotic genes are tied up with morphogenesis. We thereby describe how, by taking advantage of our imaging setup, we serendipitously uncovered that body curvature combined with in-plane tension controlled by the *Deformed* (*Dfd*) homeotic gene promotes tissue invagination.

## Results

### The invagination of the pupal Dfd homeotic domain is associated with adult neck formation

As in all Diptera, the *Drosophila* adult is characterized by a head-thorax regionalization manifested by a deep inter-segmental invagination where the neck is located[32] (Fig. 1a). In contrast, the *Drosophila* pupa does not harbor such a fold at 14 h after pupa formation (14 hAPF, Fig. 1a). To understand when and how the neck fold is formed, we performed time-lapse imaging using the E-Cadherin::3xGFP (Ecad::3xGFP) adherens junction (AJ) marker, focusing on a region covering the dorsal part of the thorax and the head (Fig. 1b). Time-lapse imaging revealed that the presumptive neck tissue starts to gradually invaginate at 18 hAPF forming a fold that progressively deepens at the head–thorax interface. This invagination is accompanied by convergent tissue flows from the head and thorax toward the presumptive neck; the head and thorax tissues thus form the flanks of the neck fold (Fig. 1b and Supplementary Movie 1). Even prior to its folding, the presumptive neck region can be identified by the presence of cells elongated along the medial-lateral (ML) axis (Fig. 1b and Supplementary Fig. 1a). We found that these cells, hereafter referred to as neck cells, are marked by the expression of the homeotic gene *Dfd*,

which also labels the adult neck (Fig. 1c, d, Supplementary Fig. 1b and Supplementary Movie 2a). To quantitatively characterize the folding dynamics, we tracked neck folding on ML transverse sections along the neck fold using Ecad::3xGFP (Fig. 1e). This enabled us to follow the successive positions of the apical fold front along the ML axis, and to precisely quantify the stereotypical tissue deepening dynamics for 8 h and up to 40 μm below the initial tissue plane (Fig. 1f, Supplementary Fig. 1c–i and Supplementary Note). We conclude that neck folding is initiated during early pupal morphogenesis by the invagination of the Dfd homeotic compartment.

### Apical and basal actomyosin-dependent regulation of in-plane ML tension controls neck folding dynamics

To investigate whether and how the homeotic Dfd compartment invagination drives neck folding, we first characterized the organization of the actomyosin cytoskeleton and the mechanical stresses in the presumptive neck region. This revealed the presence of apical and basal actomyosin structures in the neck region (Supplementary Movie 2b, c) and we further described these structures at 18 hAPF and 21 hAPF. In the apical domain, we observed the presence of supracellular apical Myosin light chain (MyoII) enrichment in the Dfd neck cells, as compared to the flanking tissues, as well as an apical actomyosin cable at the interface between the Dfd homeotic domain and the thorax cells marked by *Antennapedia* (*Antp*) expression (Fig. 2a and Supplementary Movie 2). While we imaged MyoII and F-actin along the apical-basal axis of the neck cells, we noticed that MyoII and F-actin fibers are present basally, being well aligned with the ML axis from 21 hAPF onwards (Fig. 2b and Supplementary Movie 2c). This basal

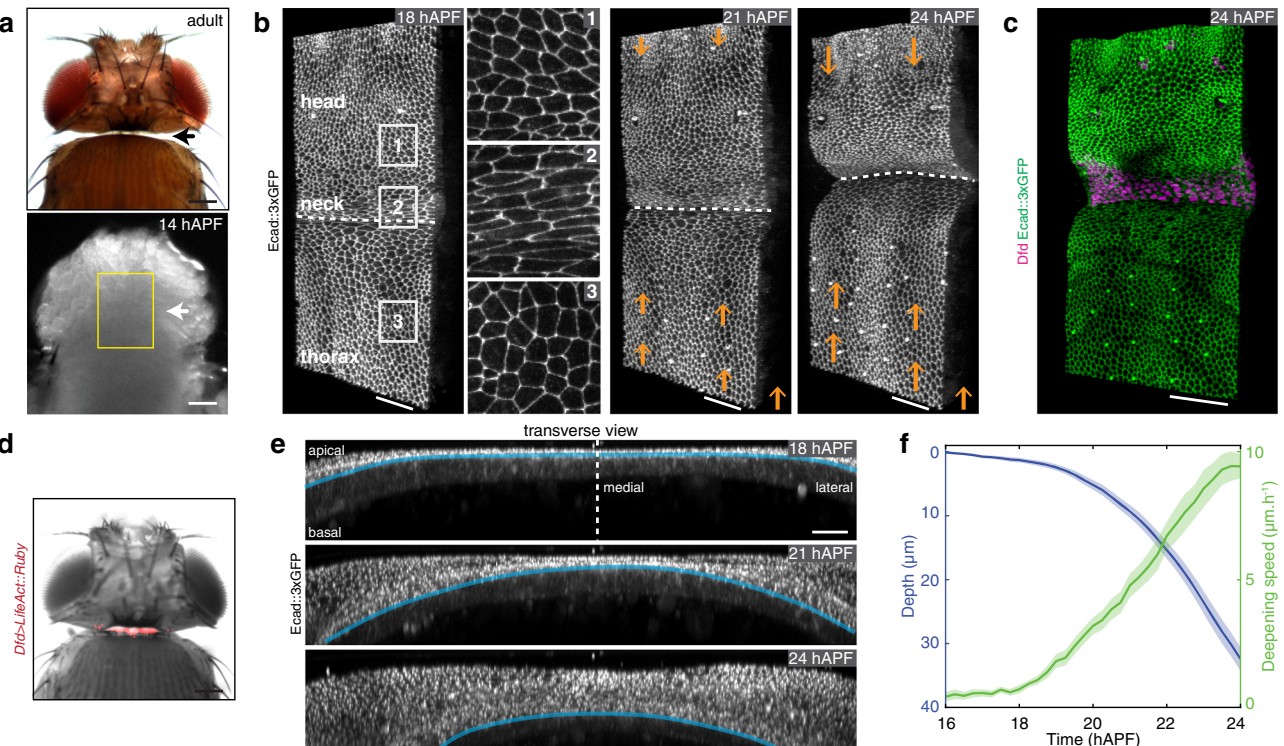

**Fig. 1 | *Drosophila* neck folding during pupal development. a** Dorsal view images of a *Drosophila* adult (top) and a 14 hAPF pupa (bottom). Arrows, the position of the adult neck. Yellow box, region imaged in (**b**). **b** Time-lapse 3D images of Ecad::3xGFP at 18, 21, and 24 hAPF during neck fold formation. See also Supplementary Movie 1. 1–3 Close-ups of the head, neck, and thorax cells in the regions indicated in the left image. Orange arrows indicate the tissue velocity in the thorax and the head from 18 to 21 hAPF and from 21 to 24 hAPF (10 μm h⁻¹, orange arrows in the bottom right). Dashed white line, apical fold front. **c** 3D image of Ecad::3xGFP and Dfd localizations in the dorsal neck region at 24 hAPF. **d** Dorsal view of a

*Drosophila* adult showing *Dfd > LifeAct::Ruby* expression in the adult neck. Note some weak signal is also detected in very small regions of the head. **e** Transverse view time-lapse images of the neck region labeled by Ecad::3xGPF at 18, 21, and 24 hAPF. Blue line, the position of the apical fold front. Dashed line, midline position. See also Supplementary Movie 1 bottom. **f** Graph of the apical neck depth (blue, mean ± sem) and of neck deepening speed (green, mean ± sem) as a function of developmental time (*N* = 10 pupae). Apical neck depth is defined relative to the initial position of the AJ labeled by Ecad::3xGFP at 16 hAPF. Source data are provided as a Source Data file. Scale bars, 50 μm (**a–d**), 20 μm (**e**).

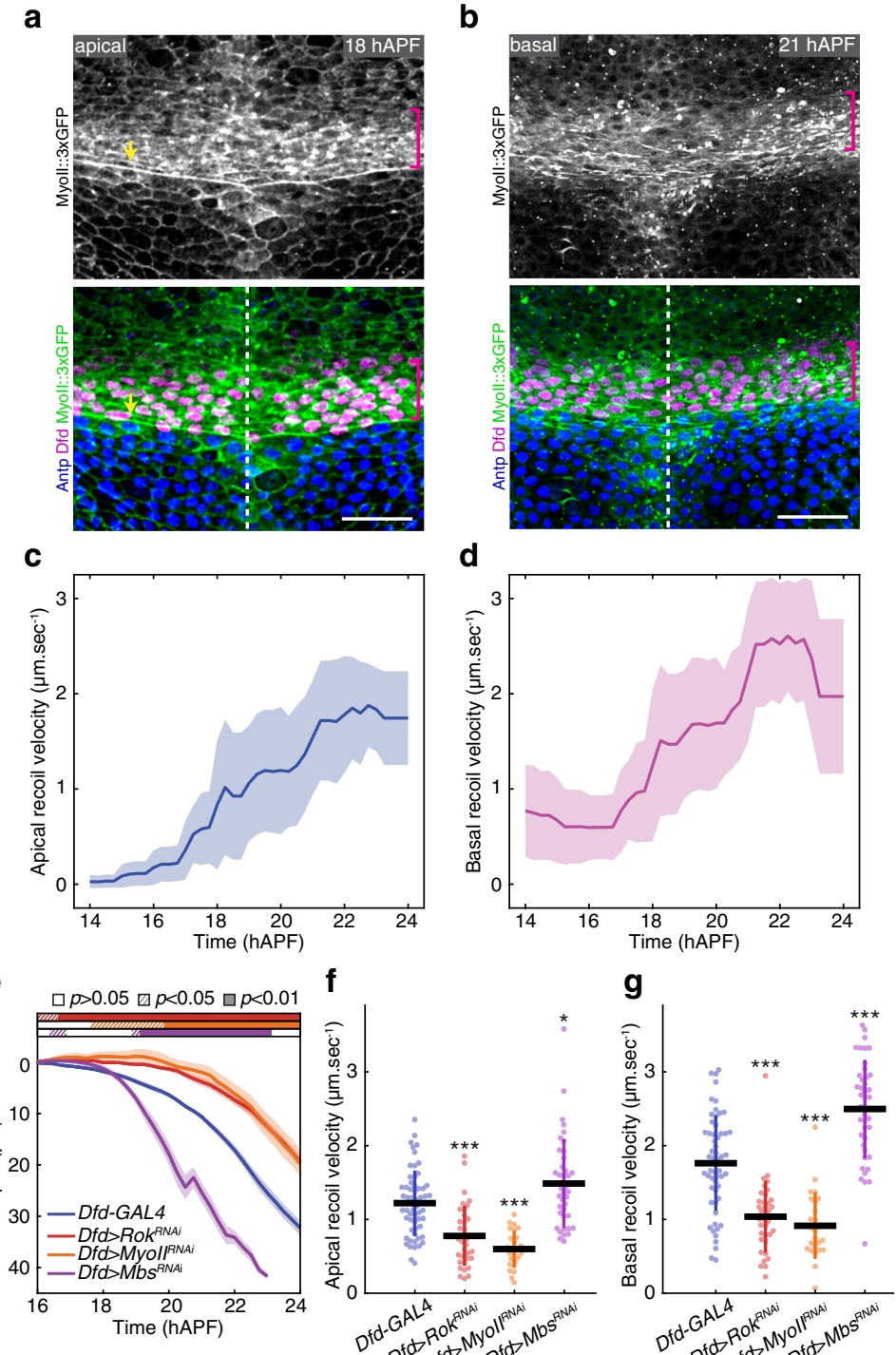

**Fig. 2 | Dfd accumulation and MyoII-dependent tension during neck folding.**
**a** Apical MyoII::3xGFP localization (gray top, green bottom) as well as Dfd (magenta bottom) and Antp (blue bottom) nuclear distributions in the neck region at 18 hAPF. Magenta brackets, neck region. Yellow arrow, apical actomyosin cable at the neck–thorax interface. Dashed line, midline position. **b** Basal MyoII::3xGFP localization (gray top, green bottom) as well as Dfd (magenta bottom) and Antp (blue bottom) nuclear distributions in the neck region at 21 hAPF. Magenta brackets, basal MyoII fibers oriented along the ML axis in the neck region. Dashed line, midline position. **c, d** Graph of the ML apical (**c**) and basal (**d**) initial recoil velocities (mean ± sd, averaged with a 2 h sliding window) upon ablation in the medial neck region as a function of developmental time ($N = 127$ pupae). See also Supplementary Movie 3. **e** Graph of the apical neck depth (mean ± sem) in control *Dfd-GAL4* ($N = 9$ pupae), *Dfd > Rok^{RNAi}* ($N = 15$ pupae), *Dfd > MyoII^{RNAi}* ($N = 7$ pupae)

and *Dfd > Mbs^{RNAi}* ($N = 13$ pupae) tissues as a function of developmental time. Horizontal boxes: *p*-values of Welch tests performed between the experimental condition and the *Dfd-GAL4* control at successive time points and color-coded according to the experimental condition (white $p > 0.05$, striped $p < 0.05$, solid $p < 0.01$). **f** Graph of the ML apical initial recoil velocities (mean ± sd) upon ablation in the neck region in control *Dfd-GAL4* ($N = 58$ pupae), *Dfd > Rok^{RNAi}* ($N = 36$ pupae, $p = 3.78e{-}06$), *Dfd > MyoII^{RNAi}* ($N = 25$ pupae, $p = 5.07e{-}12$) and *Dfd > Mbs^{RNAi}* ($N = 40$ pupae, $p = 0.02$). Ablations were performed at $20 ± 2$ hAPF. **g** Graph of the ML basal initial recoil velocities (mean ± sd) upon ablation in the neck region in control *Dfd-GAL4* ($N = 58$ pupae), *Dfd > Rok^{RNAi}* ($N = 36$ pupae, $p = 2.15e{-}08$), *Dfd > MyoII^{RNAi}* ($N = 25$ pupae, $p = 3.35e{-}09$) and *Dfd > Mbs^{RNAi}* ($N = 42$ pupae, $p = 3.72e{-}07$). Ablations were performed at $20 ± 2$ hAPF. Source data are provided as a Source Data file. Scale bars, 20 μm. Welch test, * $p < 0.05$, *** $p < 0.001$.

network is present in the Dfd neck cells and in a few cells in the most anterior region of the thorax (Fig. 2b). Having identified these distinct apical and basal supracellular actomyosin structures, we then estimated in-plane tissue stresses along the anterior-posterior (AP) and ML axes by performing multi-photon laser ablation[33] of these structures and measuring their recoil velocities (Fig. 2c, d, Supplementary Fig. 2a–f and Supplementary Movie 3). This revealed that the apical and basal actomyosin structures are under tensile stress along the ML axis, while the stress is negligible along the AP axis (Supplementary Fig. 2a). The flanking head and thorax tissues also exhibited very little AP tension, and notably much lower ML tension compared to the neck region (Supplementary Fig. 2a). Furthermore, the neck ML apical and basal recoil velocities both increased during folding and were linearly correlated (Fig. 2c, d and Supplementary Fig. 2c). In addition, the apical and basal ML recoil velocities measured in the medial versus lateral neck regions were similar (Supplementary Fig. 2d–f).

Compromising MyoII activity in neck cells by either Rho kinase dsRNA ($Dfd > Rok^{RNAi}$) or MyoII dsRNA ($Dfd > MyoII^{RNAi}$) led to slower invagination as well as reduced apical and basal recoil velocities upon ablation (Fig. 2e–g and Supplementary Fig. 2g). In addition, injection of the Rok inhibitor (Y-27623) during folding completely abolished invagination (Supplementary Fig. 2h). Conversely, increasing MyoII activity by reducing MyoII phosphatase activity ($Dfd > Mbs^{RNAi}$) caused a faster invagination and increased the apical and basal recoil velocities (Fig. 2e–g and Supplementary Fig. 2g). Taken together, our results indicate that neck folding is associated with the formation of apical and basal supracellular actomyosin structures and that MyoII activity within the Dfd domain promotes in-plane ML tension and folding dynamics.

## Dfd regulates in-plane tension via the Tollo and Dystroglycan receptors

We next aimed to identify the genetic regulators of the apical and basal actomyosin structures and mechanical stress within the Dfd homeotic compartment. Strongly reducing Dfd function in the neck ($Dfd > Dfd^{RNAi}$) lowered fold deepening while diminishing apical and basal tensions (Fig. 3a–c and Supplementary Fig. 3a, c). In addition, reduced Dfd function lowered apical MyoII levels and the alignment of the basal actomyosin network in the neck (Fig. 3d and Supplementary Fig. 3d, e). In a survey of protein localization using functional GFP-tagged gene products, we uncovered that the Toll-like receptor Tollo was enriched at the level of the AJ in the neck cells whereas the extracellular matrix (ECM) receptor Dystroglycan (Dg) and its cytoplasmic adapter Dystrophin (Dys) accumulated along the basal F-actin network (Fig. 3e, f, Supplementary Fig. 4a–c and Supplementary Movie 2d, e). Tollo and Dys/Dg control tissue elongation by regulating actomyosin organization in the *Drosophila* embryo germband and oocyte, respectively[34–36]. However, their role in tissue folding is unexplored. We found that inhibiting Tollo function in the neck ($Dfd > Tollo^{RNAi}$) slowed down neck deepening; a result that was further confirmed by analyzing *Tollo* null mutant animals (Fig. 3a and Supplementary Fig. 4d, e). Consistent with the notion that tension and deepening speed were reduced upon abrogating MyoII function, loss of Tollo function reduced apical tension and apical MyoII levels in the neck cells (Fig. 3b, g). In parallel with the analysis of Tollo function, we found that loss of Dg or Dys function (using *Dg* and *Dys* null mutant animals) as well as inhibiting Dg function in the neck ($Dfd > Dg^{RNAi}$) also slowed down folding (Fig. 4a and Supplementary Fig. 4d, f). Accordingly, the loss of Dys or Dg function compromised the basal actomyosin organization in the neck and the most anterior cells of the thorax and led to reduced basal tension (Fig. 4c–e and Supplementary Fig. 3d, e). We also analyzed neck folding in the double *Dys, Dfd > Tollo^{RNAi}* mutant context, the effect of which on invagination dynamics was similar to the one observed in $Dfd > Dfd^{RNAi}$ as well as $Dfd > Tollo^{RNAi}$ or *Dys* mutant tissues (Supplementary Fig. 5a). This prompted us to test whether Tollo and Dys/Dg might also affect basal and apical tensions, respectively. Indeed, basal recoil velocity was

reduced in $Dfd > Tollo^{RNAi}$ tissue, and conversely, Dys and Dg loss of function affected apical tension (Fig. 3c, Fig. 4b). Since Tollo is mainly localized at the apical side of the cells, independently of Dg activity, and Dg is enriched at the basal side of the cells even in the absence of Tollo activity (Supplementary Fig. 5b, c), this could suggest that apical and basal tensions feedback on each other, or that Dg and Tollo have unforeseen indirect roles on the apical and basal tension, respectively. Lastly, we found that Dfd controls both Tollo and Dg distributions in the neck compartment (Fig. 4f, g). Altogether, we propose that Dfd regulates apical and basal actomyosin organizations and in-plane ML tension via Tollo and Dys/Dg to promote tissue folding.

## An interplay between in-plane tension and curvature can account for an inward force driving folding

Our results thus far put forward the homeotic control of actomyosin organization and the resulting in-plane tension contributing to neck folding. In addition, our analysis of mechanisms previously known[7,8] to drive folding indicated that apical constriction associated with basal relaxation, apical-basal shortening, cell delamination, or tissue buckling does not substantially contribute to neck folding (Supplementary Fig. 6 and Supplementary Note). We, therefore, sought an alternative mechanism by which the in-plane ML tension in the neck region could generate a force moving cells towards the basal side of the epithelium, i.e., inside the animal. A hint was provided by the use of the coverslip needed to achieve high-resolution time-lapse imaging (Supplementary Fig. 7a, b). Without coverslip the neck region has a homogeneous curvature as observed on ML transverse section (Supplementary Fig. 7a). In contrast, the coverslip in contact with the apical ECM of the tissue locally flattened the tissue along the ML axis: the lateral regions being more curved than the medial one in contact with the coverslip (Fig. 1e and Supplementary Fig. 7b). Strikingly, while imaging with coverslip flattening, we noticed qualitatively that the lateral regions tended to invaginate faster than the flattened medial one, as was more evident at the beginning of the invagination (Supplementary Movie 4). Interestingly, these disparities of invagination dynamics were not associated with differences in apical MyoII accumulation or recoil velocities between the medial and lateral neck regions (Supplementary Fig. 7c, d and Supplementary Fig. 2d, e).

To quantitatively investigate and model the role of tissue curvature, we first used time-lapse movies to determine both the curvature and deepening speed along the ML apical fold front (Fig. 5a). Confirming our initial qualitative observations, we found that at each given developmental time during tissue invagination, the deepening speed correlates well with curvature (Fig. 5b). We, therefore, aimed to describe neck folding dynamics based on simple physical considerations regarding a curved line under tension. If the line is straight, the tension does not generate any net resulting force perpendicular to the line. If the line is curved, the symmetry is broken and the line generates a net resulting force perpendicular to it, oriented towards the curvature center (Fig. 5c). The magnitude of the resulting force, per unit length of line, is the tension **T** multiplied by the curvature $\kappa$, also called "Laplace" force; it has been applied previously to the one-dimensional border of a two-dimensional epithelium when modeling epithelium closure[37–41]. Here, since the tissue thickness (~10 μm) is smaller than the neck radius of curvature (~300 μm), we applied the equation of Laplace force to a thin epithelial strip that moves inwards perpendicular to its surface (see Supplementary Note). Inertia being negligible with respect to viscous energy dissipation, we write that the sum of this Laplace force and of the normal dissipative force is zero; the latter force being equal to the deepening speed (velocity $\mathbf{v_n}$ of the line along the vector normal to the curve) multiplied by an unknown negative dissipative prefactor, $-\mu$. So, the prediction is that $\kappa \mathbf{T} - \mu \mathbf{v_n} = 0$ or equivalently, that $\mathbf{v_n}$ is proportional to $\kappa \mathbf{T}$ and is directed inwards. To experimentally test this prediction, we computed the product of the tissue ML tension and the apical fold front local curvature and then compared it

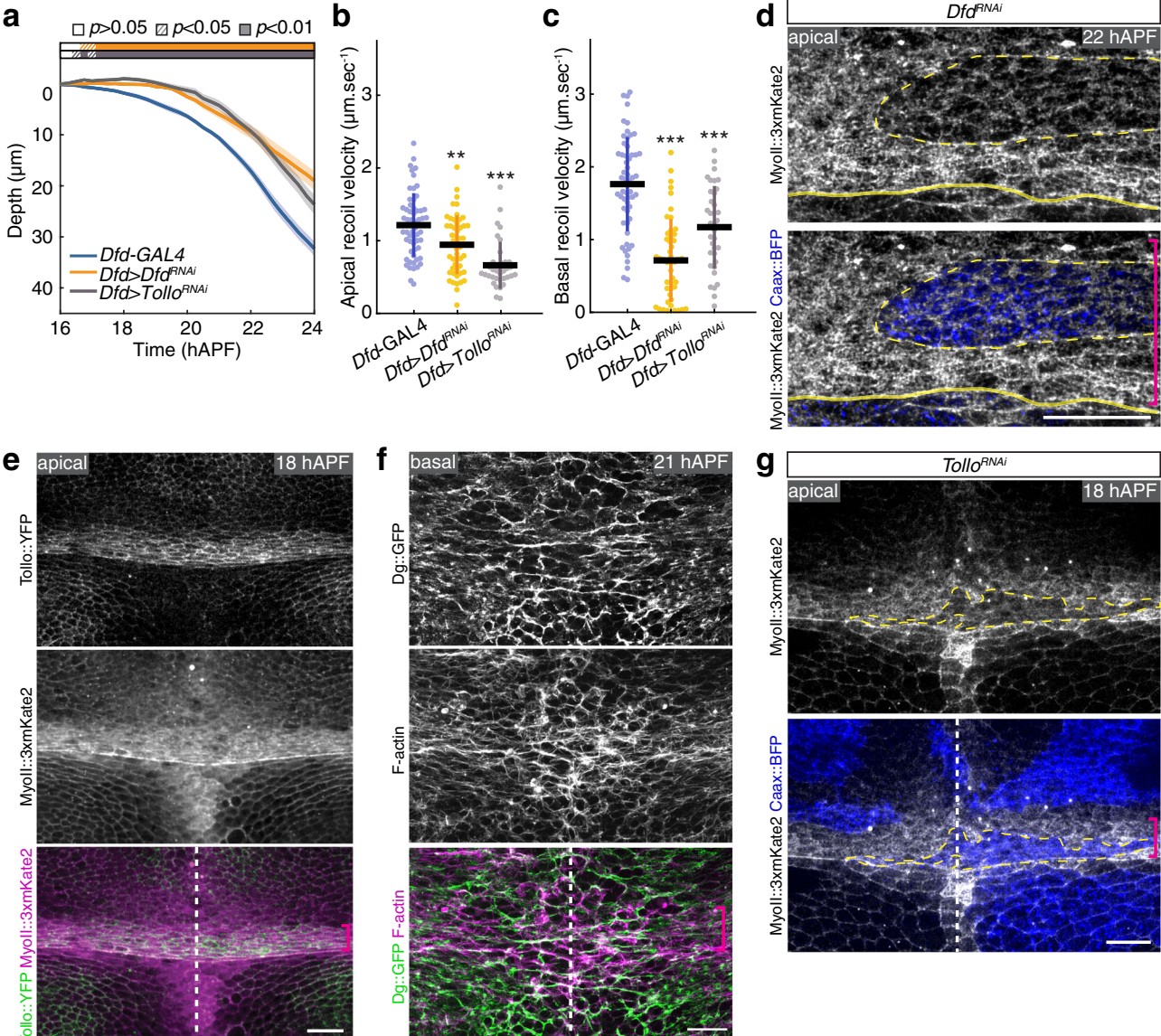

**Fig. 3 | Control of ML in-plane tissue tension by Dfd and Tollo during neck folding. a** Graph of the apical neck depth (mean ± sem) in control *Dfd-GAL4* (*N* = 9 pupae), *Dfd > Dfd^RNAi* (*N* = 8 pupae) and *Dfd > Tollo^RNAi* (*N* = 11 pupae) tissues as a function of developmental time. Horizontal boxes: *p*-values of Welch tests performed between the experimental condition and the *Dfd-Gal4* control at successive time points and color-coded according to the experimental condition (white *p* > 0.05, striped *p* < 0.05, solid *p* < 0.01). **b** Graph of the ML apical initial recoil velocities (mean ± sd) upon ablation in the neck region in control *Dfd-GAL4* (*N* = 58 pupae), *Dfd > Dfd^RNAi* (*N* = 49 pupae, *p* = 1.33e−3) and *Dfd > Tollo^RNAi* (*N* = 36 pupae, *p* = 4.97e−10). Ablations were performed at 20 ± 2 hAPF. **c** Graph of the ML basal initial recoil velocities (mean ± sd) upon ablation in the neck region in control *Dfd-GAL4* (*N* = 58 pupae), *Dfd > Dfd^RNAi* (*N* = 49 pupae, *p* = 2.20e−14) and *Dfd > Tollo^RNAi* (*N* = 36 pupae, *p* = 1.00e−5). Ablations were performed at 20 ± 2 hAPF. **d** Apical

MyoII::3xmKate2 distribution in *Dfd^RNAi* clones marked by Caax::BFP at 22 hAPF in the neck region. Yellow dashed lines, outlines of *Dg^RNAi* clone in the neck region. Magenta bracket, neck region. Yellow line, neck–thorax boundary. **e** Apical Tollo::YFP (gray top, green bottom) and MyoII::3xmKate2 (gray center, magenta bottom) distributions at 18 hAPF in the neck region. Magenta bracket, neck region. Dashed line, midline position. **f** Basal Dg::GFP (gray top, green bottom) and F-actin (gray center, magenta bottom) distributions at 21 hAPF in the neck region. Magenta bracket, neck region. Dashed line, midline position. **g** Apical MyoII::3xmKate2 distribution in *Tollo^RNAi* clones marked by Caax::BFP at 18 hAPF in the neck region. Yellow dashed lines, outlines of *Tollo^RNAi* clone in the neck region. Magenta bracket, neck region. White dashed line, midline position. Source data are provided as a Source Data file. Scale bars: 10 μm (**d**), 20 μm (**e–g**). Welch test, **p* < 0.005, ***p* < 0.001.

---

to the local deepening speed. Strikingly, we found that the data of folding dynamics at different developmental times now collapse along a single straight line showing a linear relationship with high correlation between the deepening speed on the one hand, and the product of the apical fold front curvature and tissue ML tension on the other hand (Fig. 5d, $R^2 = 0.96$). Interestingly, this proportionality relationship implies that the local modulation of neck folding dynamics with coverslip flattening is unlikely to be explained by changes in the friction or the adhesion between the apical ECM and the epithelium in the regions flattened by the coverslip (see Supplementary Note). Other mechanical

changes indirectly induced by the flattening are equally unlikely due to the small magnitude of the resulting compression (see Supplementary Note). Importantly, we also assessed whether deepening speed is well predicted in experimental conditions where ML tension is modulated. We observed that in *Dfd > Rok^RNAi*, *Dfd > MyoII^RNAi*, *Dfd > Dfd^RNAi*, *Dfd > Tollo^RNAi*, *Dys*, or *Dg* mutant conditions for which the tensions were decreased, the deepening speed again linearly correlated with the product of the estimated tension and the local curvature (Fig. 5e, f, Supplementary Fig. 8 and Supplementary Note); this also holds true when increasing MyoII activity in *Dfd > Mbs^RNAi* animals. Again, the data

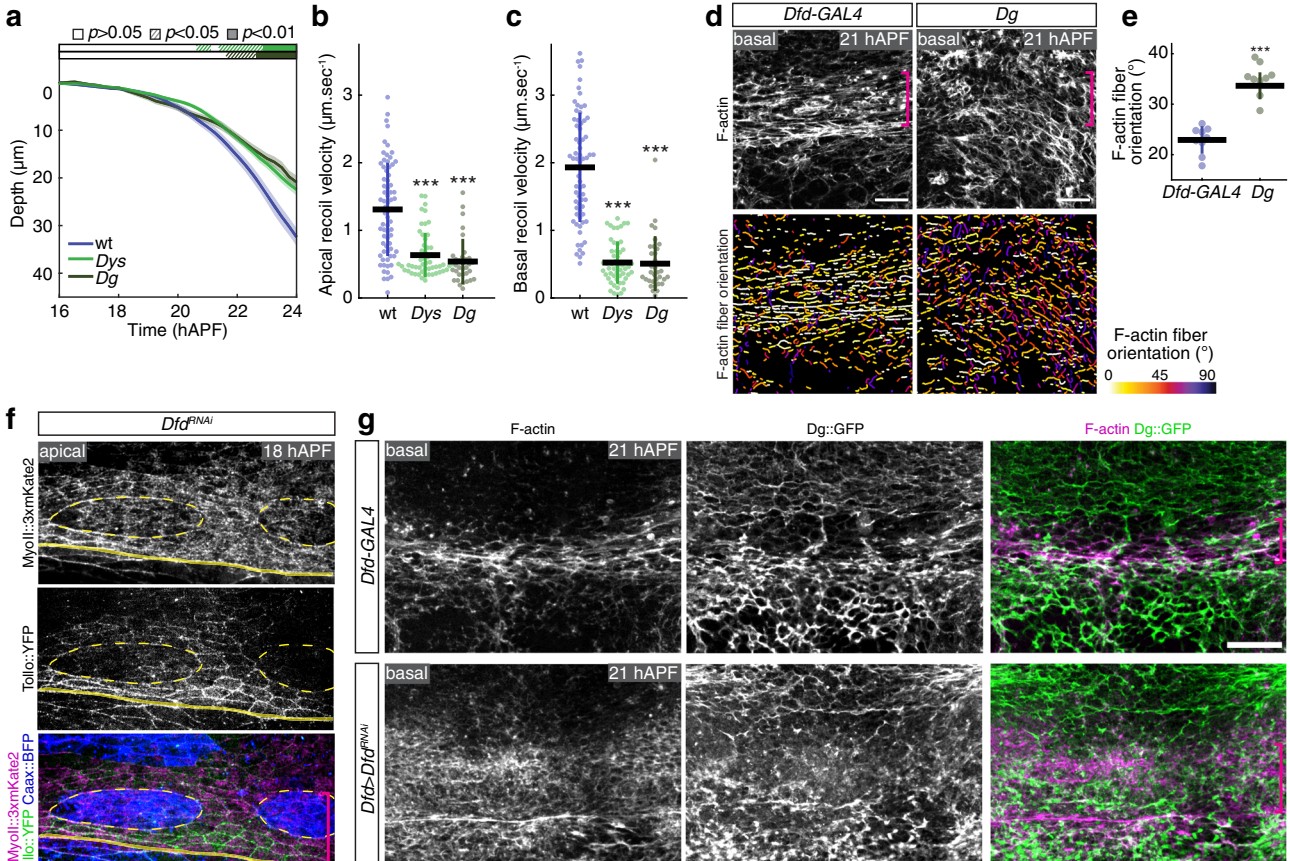

**Fig. 4 | Control of ML in-plane tissue tension by Dys/Dg during neck folding.**
**a** Graph of the apical neck depth (mean ± sem) in wt ($N = 10$ pupae), Dys ($N = 12$ pupae), and Dg ($N = 10$ pupae) tissues as a function of developmental time. Horizontal boxes: p-values of Welch tests performed between the experimental condition and the wt control at successive time points and color-coded according to the experimental condition (white $p > 0.05$, striped $p < 0.05$, solid $p < 0.01$). **b** Graph of the ML apical initial recoil velocities (mean ± sd) upon ablation in the neck region in wt ($N = 69$ pupae), Dys ($N = 53$ pupae, $p = 1.36e{-}10$) and Dg ($N = 33$ pupae, $p = 1.72e{-}11$) tissues. Ablations were performed at $20 ± 2$ hAPF. t-test, ***$p < 0.001$. **c** Graph of the ML basal initial recoil velocities (mean ± sd) upon ablation in the neck region in wt ($N = 69$ pupae), Dys ($N = 53$ pupae, $p = 4.54e{-}23$), and Dg ($N = 33$ pupae, $p = 9.13e{-}21$) tissues. Ablations were performed at $20 ± 2$ hAPF. **d** Basal F-actin distribution (top) and segmentation of the F-actin fibers color-coded by orientation (bottom) at 21 hAPF in control Dfd-GAL4 and Dg tissues. Segmented F-actin fibers are colored according to their orientation, ranging from 0° (along the

ML axis) to 90° (along the AP axis). While F-Actin fiber orientation is affected in Dg mutant tissues, the phalloidin signal level is similar in control and Dg mutant tissues. Magenta brackets, neck region. **e** Graph of the F-actin fibers orientation relative to the ML axis (mean ± sd) in control Dfd-GAL4 ($N = 9$ pupae) and Dg ($N = 9$ pupae) tissues in the neck region at 21 hAPF. Each F-actin fiber orientation is measured between 0° and 90° relative to the ML axis (i.e., 0° corresponding to a fiber parallel to the ML axis). Each dot corresponds to the fiber orientation value averaged for one animal. Welch test, $p = 2.66e{-}07$. **f** Apical MyoII::3xmKate2 (gray top, magenta bottom) and Tollo::YFP (gray center, green bottom) distributions in $Dfd^{RNAi}$ clones marked by Caax::BFP accumulation (blue bottom) at 18 hAPF. Yellow dashed lines, outlines of $Dfd^{RNAi}$ clones in the neck region. Magenta bracket, neck region. Yellow line, neck–thorax boundary. **g** Basal F-actin (gray left, magenta right) and Dg::GFP (gray center, green right) distributions in control Dfd > GAL4 (top) and Dfd > $Dfd^{RNAi}$ (bottom) tissues at 21 hAPF. Magenta brackets, neck region. Source data are provided as a Source Data file. Scale bars: 20 μm. Welch test, ***$p < 0.001$.

---

of folding dynamics collapse along a single straight line (Fig. 5e, f, Supplementary Fig. 8b). We note that the slope of the linear relationship between the deepening speed and the product of estimated tissue ML tension and curvature varies between wild-type (wt) and some mutant conditions (Supplementary Fig. 8c). Since we assessed tissue tension via recoil velocity that depends upon tension as well as tissue mechanical properties (friction and viscosity)[33,42], the slope variabilities may indicate that some mutant conditions could also affect tissue mechanical properties. This is for example the case for Dg loss of function exhibiting a higher slope than the one obtained in wt conditions (Supplementary Fig. 8c), as it promotes a strong decrease of recoil velocity while having a more modest impact on invagination dynamics (Fig. 4a, b). Last, while basal curvature measurements were less accurate, we also found a linear relationship between the deepening speed and the product of basal fold front curvature and ML tension on the basal fold front (Supplementary Figs. 9 and 10). Altogether, these results agree with a model in which the product of curvature and tension can account for neck folding dynamics.

## Tissue curvature regulates the spatiotemporal dynamics of folding

Having investigated the role of in-plane tension, we next challenged the role of curvature by testing whether modulating the local tissue curvature would result in changes in deepening speed. Toward this goal, we developed a method to image the animal with a coverslip, but without inducing flattening, while maintaining a reasonable signal-to-noise ratio to track Ecad::3xGFP AJ signal (Fig. 6a and Supplementary Movie 4). In conditions without flattening, the neck apical fold front shows a homogeneous and high curvature prior to folding. As with flattened controls, apical and basal tensions increase over time in non-flattened animals (Supplementary Fig. 11a, b). In agreement with the role of curvature in defining the resulting inward forces, we found that (i) tissue deepening speed is homogeneous along the ML axis in these animals, correlating with homogenously high curvature; (ii) despite the lower apical recoil velocities relative to the ones measured in flattened conditions, tissue deepening occurs earlier and faster in non-flattened animals (Fig. 6b–d and Supplementary Fig. 11a and

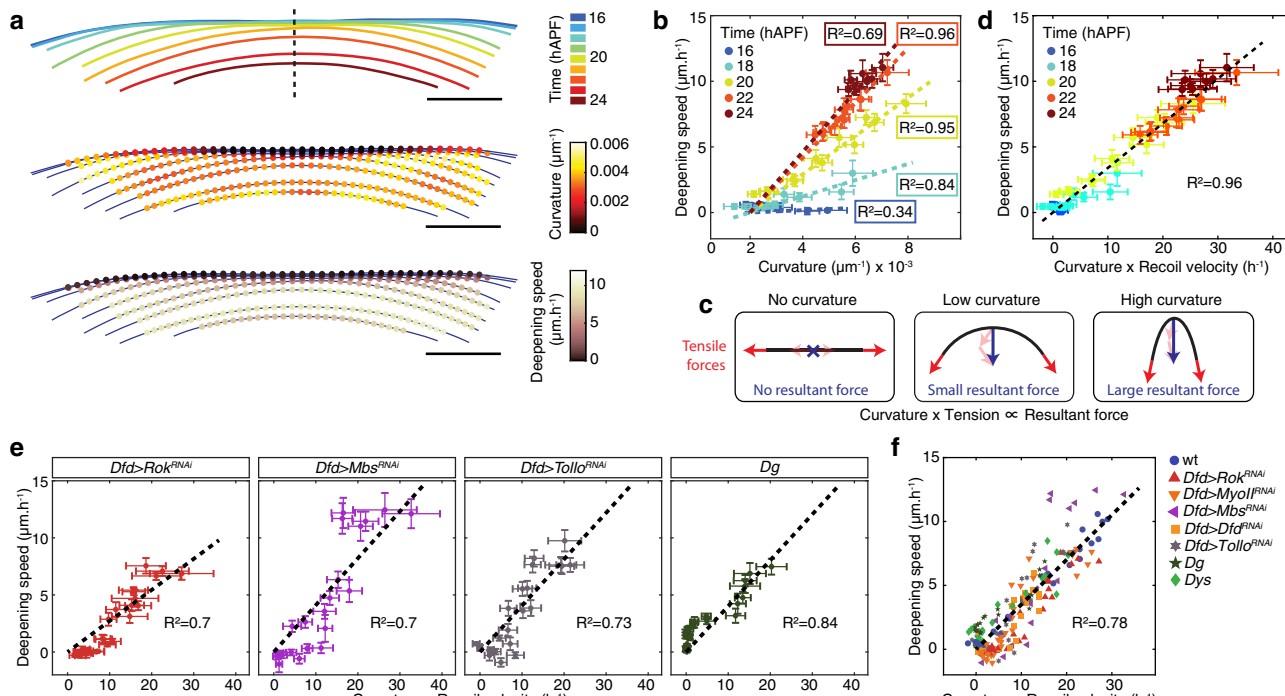

**Fig. 5 | The product of local tissue curvature and ML tension can account for neck folding dynamics. a** Successive positions of the apical fold front, color-coded according to time (top), local tissue curvature (middle), and local deepening speed (velocity along the vector normal to the curve) of the apical fold front (bottom) for an individual pupa. Dashed line, midline position. See also Supplementary Movie 4. **b** Graph of the apical fold front deepening speed (mean ± sem) versus curvature (mean ± sem). Each point is the mean value (± sem) among 10 control animals for a given Position$_{ML}$ at a given color-coded developmental time. For each developmental time, a line passing through the origin was fitted ($R^2$ values for each developmental time are indicated). **c** Schematic illustrating that the tissue line tension and the local tissue curvature define the magnitude of the normal resulting force (see text and Supplementary Note for details). **d** Graph of the apical fold front deepening speed (mean ± sem) and of the product of curvature and ML apical initial recoil velocity upon laser ablation in the neck region (mean ± sem) for a given Position$_{ML}$ at a color-coded developmental time. Datasets from Fig. 1f and Fig. 2c were used (see corresponding figure legends for sample size). A line passing through the origin was fitted to the average values ($R^2 = 0.96$). **e, f** Graph of the apical fold front deepening speed (mean ± sem) versus the product of curvature and ML apical initial recoil velocity upon laser ablation in the neck region (mean ± sem) for a given Position$_{ML}$ and selected genotypes: *Dfd > Rok$^{RNAi}$*, *Dfd > Mbs$^{RNAi}$*, *Dfd > Tollo$^{RNAi}$*, and *Dg* (**e**), or all tested experimental conditions (**f**). Datasets from Fig. 2e, Fig. 3a, Fig. 4a and Supplementary Fig. 8a were used (see corresponding figure legends for sample size). A line passing through the origin was fitted (black dashed line, $R^2$ values are indicated). Source data are provided as a Source Data file.

Supplementary Movie 4). Furthermore, the deepening speed is linearly correlated with the product of tension and curvature, suggesting that coverslip flattening modulates the invagination dynamics per se, rather than the underlying mechanisms (Fig. 6e and Supplementary Fig. 11c, d). If curvature modulates folding dynamics, reducing curvature in any region of the fold should locally reduce deepening speed. We thus flattened the lateral side of the neck region by mounting the pupa on its side; and used the contralateral non-flattened side as an internal control (Fig. 6f and Supplementary Movie 4). In these experimental conditions, the flattened lateral side behaves like animals imaged with coverslip flattening in the medial domain, while the control non-flattened lateral side deepens with faster dynamics (Fig. 6g–i and Supplementary Fig. 11c). Altogether, we conclude that modulating the curvature led to changes in deepening speed that is predicted by the product of the curvature and the tension (Fig. 6e, j). Interestingly, we also noticed that independently of the initial curvature heterogeneity imposed by the coverslip, the tissue shape, with flat and curved regions, converged to a more homogenous curved geometry during invagination (Fig. 5a, Fig. 6g and Supplementary Movie 4). In fact, since the velocity is proportional to curvature, a more curved region tends to shrink more quickly (Fig. 5a), thus with time the curvature increases and becomes more homogeneous (Supplementary Fig. 11e and Supplementary Note). This purely geometrical property may therefore account for a robust neck invagination process, independent of the initial heterogeneities in curvature due to our imaging set-up.

Having found that Dfd homeotic compartment curvature impacts the spatiotemporal dynamics of neck invagination, we aimed at analyzing whether the presence of a curved region is necessary to promote folding. Prior to neck folding, we used a UV laser to ablate the tissue across its whole thickness to mechanically isolate the medial region from the lateral curved regions (Supplementary Fig. 11f). We then imaged neck folding while repeating the tissue severing ablations every 3 h (Fig. 7 and Supplementary Movie 5). When the isolated medial region was initially flattened by the coverslip, the tissue deepening speed was drastically reduced, while the tissue remained under ML tension (Fig. 7a–c and Supplementary Fig. 11g, h). Strikingly, without flattening, the isolated curved tissue initially deepened, and as it became flatter its deepening speed reduced (Fig. 7d–f). These experiments agree with the hypothesis that an initial curvature within the fold plane promotes folding and that a flat region can deepen if mechanically coupled to a curved one; thus, confirming the role of curvature in controlling the spatiotemporal dynamics of neck folding.

## Discussion

The morphogenetic and functional specification of tissue compartments is a major trait of animal development. As such, the exploration of tissue morphogenesis and homeotic gene function is key to unraveling the conserved mechanisms driving development. Here we uncovered that the function of Dfd homeotic gene in folding morphogenesis can be understood by integrating its activity with the initial animal geometry upon which it acts on. Accordingly, we propose that

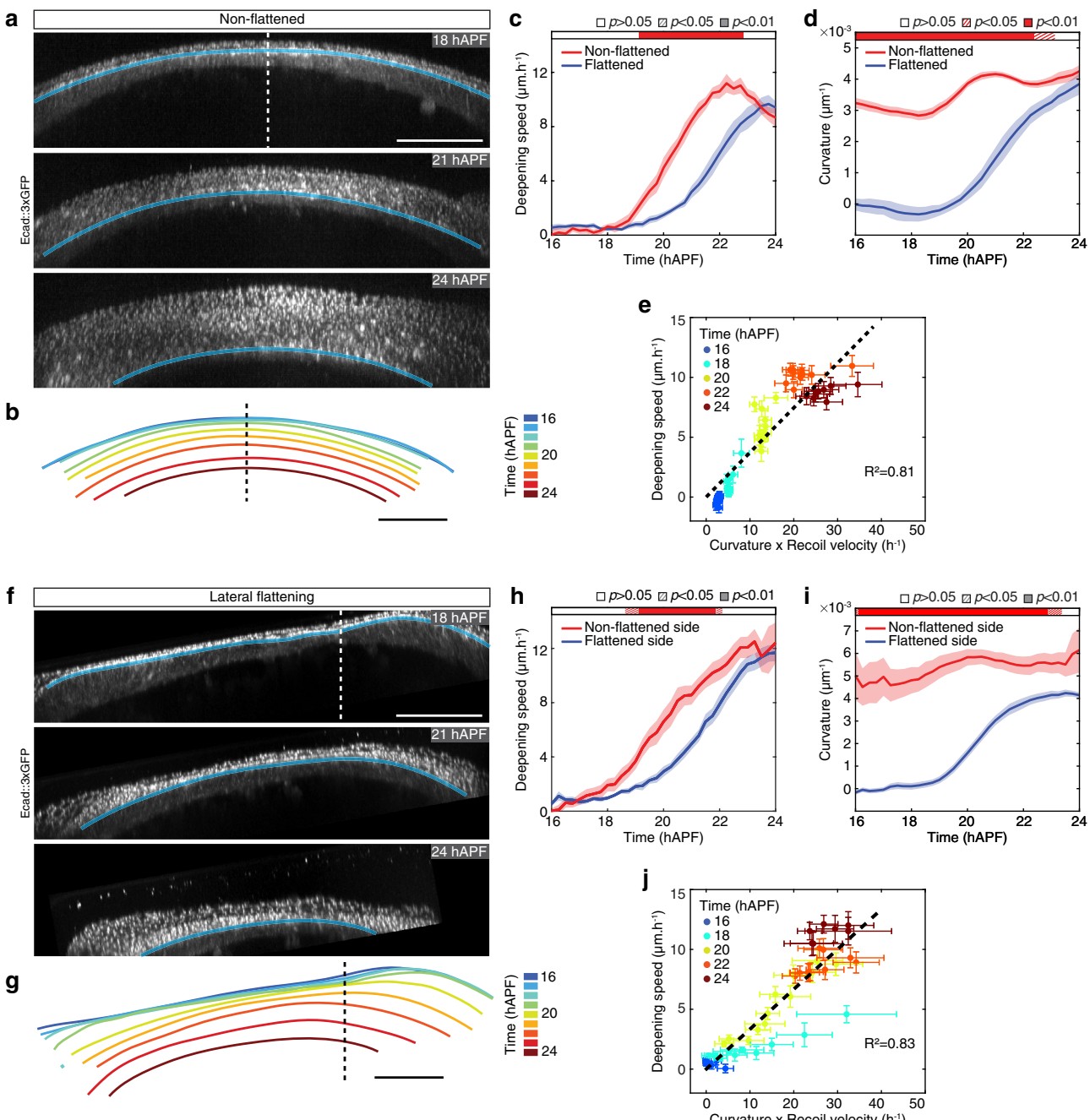

the establishment of in-plane tension combined with an initially convex curved geometry promotes an inward force controlling folding dynamics (Fig. 8). Importantly, such spatiotemporal regulation by the product of tension and tissue curvature distinguishes this folding mechanism from known ones and defines a central role of tissue curvature in morphogenesis. While future work will be necessary to delineate how Dfd controls Tollo and Dg distributions and whether additional regulators act downstream of Dfd (Fig. 8), the interplay between tissue tension and curvature likely expands beyond the fold we have explored here. Indeed, epithelial tissues in vivo are closed surfaces that by default harbor non-zero convex curvature and are often under tensile stress[7,43,44]. In particular, many tubular organs and curved embryonic epithelia are physically compartmentalized by transverse folds[3-6,22], calling for further analyses of the roles of tissue and animal curvature during such fold formation. By linking in-plane tension and tissue curvature our work more generally helps to understand the role of tissue geometry in biology.

In addition to uncovering the critical role of tissue geometry in folding dynamics, our findings raise several important questions in the field of tissue morphogenesis. A distinctive feature of the folding process we have characterized is that the apical and basal tissue tensions are highly correlated. We found that apical Tollo and basal Dg regulate basal and apical tensions without controlling each other's localizations and we did not detect any short time scale (tens of seconds) mechanical coupling between apical and basal tensions (Supplementary Fig. 6i, j). Therefore, we envision that the correlated apical and basal tensions are more likely linked to long-timescale mechanical feedback or to an indirect role of Tollo or Dg on the basal or apical tension, respectively (Fig. 8), via for example the apical-basal microtubule networks. The characterization of the putative mechanical feedback and the potential indirect roles of Tollo and Dg will be important to better understand how tissue folding and tissue thickness are controlled during neck morphogenesis, and more generally how the crosstalk between apical and basal tension is regulated in epithelial

**Fig. 6 | Local curvature along the neck fold controls folding dynamics.**
**a** Transverse view time-lapse images of the neck region labeled by Ecad::3×GPF at 18, 21, and 24 hAPF in a non-flattened pupa (i.e., imaged without coverslip flattening). Blue line, the position of the apical fold front. Dashed line, midline position. See also Supplementary Movie 4. **b** Successive positions of the apical fold front, color-coded according to time, in a pupa imaged without coverslip flattening (**a**). Dashed line, midline position. **c** Graph of apical fold front deepening speed (mean ± sem) in pupae imaged without coverslip flattening (non-flattened, $N = 11$ pupae) and with coverslip flattening (flattened, $N = 101$ pupae) as a function of developmental time. Horizontal box: $p$-values of Welch tests performed between non-flattened and flattened experimental conditions at successive time points (white $p > 0.05$, striped $p < 0.05$, solid $p < 0.01$). **d** Graph of curvature (mean ± sem) in pupae imaged without coverslip flattening (non-flattened, $N = 110$ pupae) and with coverslip flattening (flattened, $N = 10$ pupae) as a function of developmental time. Horizontal box: $p$-values of Welch tests performed between non-flattened and flattened experimental conditions at successive time points (white $p > 0.05$, striped $p < 0.05$, solid $p < 0.01$). **e** Graph of the apical fold front deepening speed (mean ± sem) and of the product of curvature and ML apical initial recoil velocity upon laser ablation in the neck region (mean ± sem) in pupae imaged without coverslip flattening for a given Position$_{ML}$ at a given color-coded developmental time. Datasets from Fig. 6c, d and Supplementary Fig. 11a were used (see corresponding figure legends for sample size). A line passing through the origin was fitted ($R^2 = 0.81$). **f** Transverse view time-lapse images of the neck region labeled by

Ecad::3×GPF at 18, 21, and 24 hAPF in a pupa flattened laterally relative to the midline. Blue line, the position of the apical fold front. Dashed line, midline position. **g** Successive positions of the apical fold front, color-coded according to time, in a pupa laterally flattened (**f**). Dashed line, midline position. See also Supplementary Movie 4. **h** Graph of apical fold front deepening speed (mean ± sem) of the non-flattened side and flattened side in pupae laterally flattened relative to the midline (as in (**f**), $N = 11$) as a function of developmental time. Horizontal box: $p$-values of Welch tests performed between the non-flattened side and flattened side of the same pupae at successive time points (white $p > 0.05$, striped $p < 0.05$, solid $p < 0.01$). **i** Graph of curvature (mean ± sem) of the non-flattened side and flattened side in pupae flattened laterally relative to the midline (as in (**f**), $N = 11$) as a function of developmental time. Horizontal box: $p$-values of Welch tests performed between the non-flattened side and flattened side of the same pupae at successive time points (white $p > 0.05$, striped $p < 0.05$, solid $p < 0.01$). **j** Graph of the apical fold front deepening speed (mean ± sem) and of the product of curvature and ML apical initial recoil velocity upon laser ablation in the neck region (mean ± sem) in laterally flattened pupae for a given Position$_{ML}$ at a given color-coded developmental time. Recoil velocities measured in flattened animals (Fig. 2c, see corresponding figure legends for sample size) were used for regions of initial curvatures below a threshold of 0.00015 µm$^{-1}$, while recoil velocities measured in non-flattened animals (Supplementary Fig. 11a) were used for regions of initial curvatures larger than 0.00015 µm$^{-1}$. A line passing through the origin was fitted ($R^2 = 0.83$). Source data are provided as a Source Data file. Scale bars, 50 µm.

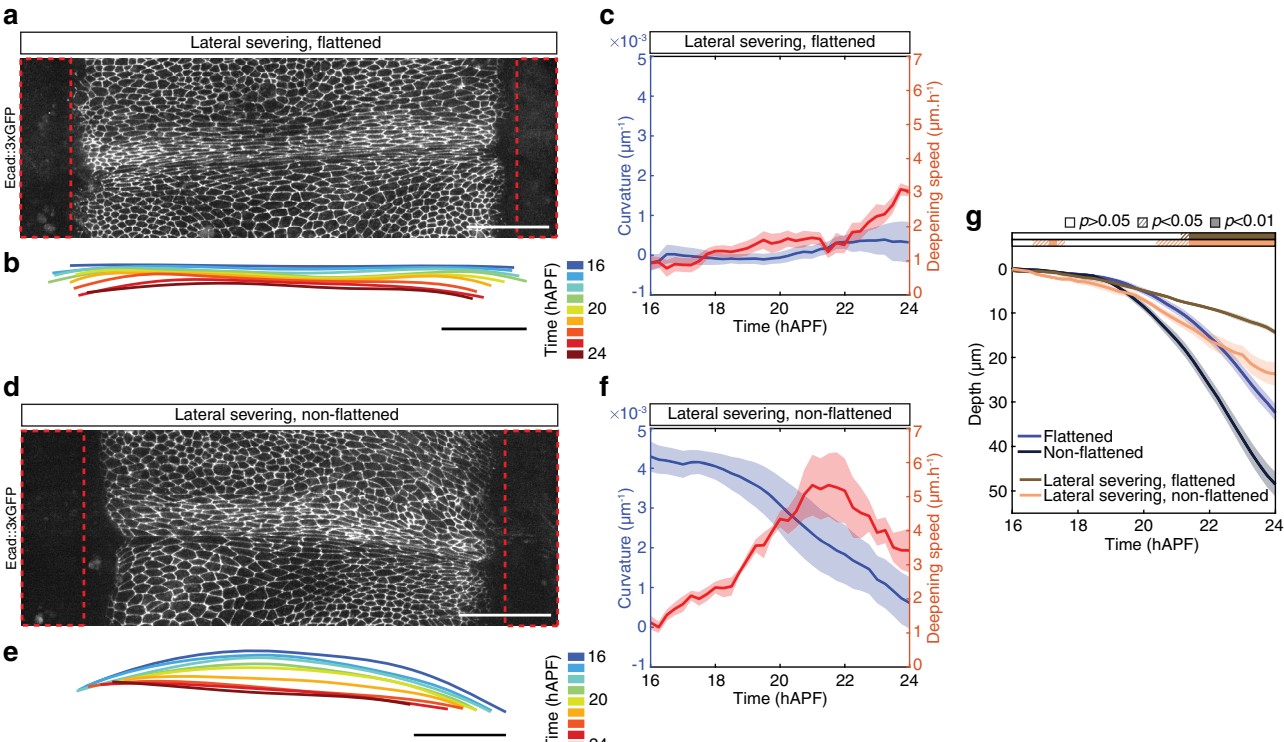

**Fig. 7 | Local curvature is necessary for neck invagination. a, b** Dorsal view image of Ecad::3xGFP distribution upon tissue severing ablation of each lateral side of the neck (red dashed boxes) (**a**) and corresponding successive positions (color-coded according to time) of the apical fold front in transverse views (**b**) in a pupa mounted with medial coverslip flattening. See also Supplementary Movie 5. **c** Graph of curvature (blue, mean ± sem) and apical fold front deepening speed (red, mean ± sem) in pupae with two lateral side tissue severing ablations and imaged with medial coverslip flattening ($N = 9$ pupae) as a function of developmental time. **d, e** Dorsal view image of Ecad::3xGFP distribution upon tissue severing ablation of each lateral side of the neck (red dashed boxes) (**d**) and corresponding color-coded successive positions of the apical fold front in transverse views (**e**) in a pupa mounted without coverslip flattening (non-flattened). See also Supplementary Movie 5. **f** Graph of

curvature (blue, mean ± sem) and apical fold front deepening speed (red, mean ± sem) in pupae with lateral side tissue severing ablations and imaged without coverslip flattening ($N = 6$ pupae) as a function of developmental time. **g** Graph of apical neck depth (mean ± sem) in pupae without lateral side ablation and imaged with (flattened, $N = 10$ pupae) and without (non-flattened, $N = 11$ pupae) coverslip flattening as well as in pupae with two lateral side ablations and imaged with (flattened, $N = 9$ pupae) and without coverslip flattening (non-flattened, $N = 6$ pupae) as a function of developmental time. Horizontal boxes: $p$-values of Welch tests performed between the flattened and ablated, flattened condition (brown) as well as between the non-flattened and ablated, non-flattened condition (orange) at successive time points (white $p > 0.05$, striped $p < 0.05$, solid $p < 0.01$). Source data are provided as a Source Data file. Scale bars, 50 µm.

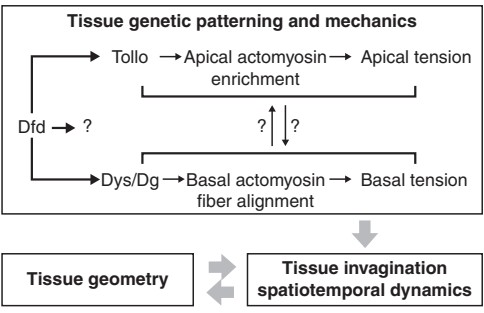

**Fig. 8 | Proposed model for the interplay between genetic patterning, tissue geometry, and mechanical tension in the control of tissue folding.** Genetic patterning leads to the establishment of in-plane ML apical and basal tension by Dfd-dependent regulation of the apical actomyosin enrichment and basal F-actin fiber organization via Tollo and Dys/Dg. Dfd might regulate Tollo and Dg accumulation via either similar or distinct mechanisms and our data do not exclude that additional Dfd downstream effectors modulate ML tension to promote neck invagination. The existence of putative feedback between apical and basal tension is indicated by question marks (see section "Discussion").

tissues. We note that we have used laser ablation to probe mechanical tension as validated in multiple experimental contexts[33,42]. Therefore, it remains difficult to fully explain all temporal variations and differences in apical and basal recoil velocities as they could reflect modifications in the apical versus basal tissue mechanical properties, due to distinct friction on the apical versus basal ECM, for example. Besides, future works will be necessary to apprehend how apical and basal tension are temporally controlled. An additional feature unveiled by our work relates to the presence of two types of supracellular actomyosin structures in the apical region: the apical actomyosin cable at the boundary between the neck and thorax and the strong actomyosin enrichment in the neck cells. We did not aim at separating the contributions of these two supracellular structures and instead targeted deciphering the role of tissue curvature. Indeed, our theoretical model used to explore the role of tissue geometry gives very good predictions when treating the neck as a line and measuring recoil velocities across the two structures. We foresee that the characterization of the putative respective roles of these two actomyosin structures could help to delineate how (i) the boundary between homeotic compartments is controlled, (ii) the detailed sagittal shape of the invagination is modulated, and (iii) the feedback between apical and basal tensions is controlled. Finally, a last feature relates to the large scale of the neck fold we studied, and thus, the permissive versus active contributions of the neighboring tissue flows. Since the neck fold is very large relative to the size of the cells, the head and thorax tissue flows are substantial relative to previous folding processes driven by apical constriction[7,8]. Our ablation of head and thorax tissues illustrated that on the time-scale of 5–6 h (Supplementary Fig. 6n, o and Supplementary Note), neck folding is not driven by tissue buckling due to the opposite cell flows of these two tissues, agreeing with a permissive role of the head and thorax flows. On a longer time scale, around 6 h post ablation, the deepening speed started to decrease relative to non-ablated control tissue, presumably as a long-term consequence of the tissue stretching observed in the head and thorax cells near the ablation sites (Supplementary Fig. 6n). In the future it will be interesting to explore whether over long time scales surrounding tissue flows contribute to large fold formation during development. Altogether, by exploring large fold formation, we have uncovered how the interplay between in-plane tension and tissue curvature provides a simple, robust, and possibly a general mechanism for fold formation. In addition, our work raises novel questions in the control of 3D tissue morphogenesis by coupled apical and basal tensions, distinct supracellular actomyosin structures, and large-scale tissue flows.

So far, in biological systems, the curvature has been proposed to play a fundamental role at the subcellular scale in membranes and cytoskeleton dynamics and the role of curvature has been analyzed during cytokinesis, tissue spreading or flows, tumor growth, and cell fate specification[37,41,45–49]. However, the role of curvature in controlling the spatiotemporal dynamics of folding has remained uncharacterized despite theoretical work[50,51]. Our work also extends the study of supracellular actomyosin structures in tissue development and repair[30,37,39,41,52–55] by defining how their curvature imposed by animal geometry impinges on the spatiotemporal dynamics of out-of-plane deformations. So far, buckling instabilities driven by tissue compression have been shown on several examples to explain large fold formation, with regional differences in growth or geometry defining their position[25–27,56,57]. Our study exemplifies how in-plane tension combined with tissue compartment geometry induces a large fold at a genetically defined position independently of mechanical instabilities. It further suggests that simple feedback between local tissue geometry and folding velocity can buffer initial spatial variations in tissue geometry. Our findings therefore complement previous results linking folding robustness to the exquisite molecular control of actomyosin dynamics or organization[21,58,59]. Collectively our work highlights a simple design principle driving fold formation and more generally calls for a better understanding of the role of tissue geometry in morphogenesis during development.

## Methods

### Fly stocks and genetics

Supplementary Table 1 lists *Drosophila melanogaster* stocks used and associated references. Loss-of-function and gain-of-function experiments were carried out using the Gal4/UAS system utilizing the temperature-sensitive Gal80[ts60,61], except for the *Dfd > Dfd^RNAi* and *Dfd > Dg^RNAi* experiments. To tune the transcriptional activity of the *Dfd-GAL4* or *Act-GAL4* drivers, animals were raised at 18 °C and incubated at 29 °C for 48 h prior to imaging or for 8 h in the case of *Dfd > Diap1*. As in any temperature shift experiment using the Gal4 system, the magnitude of the loss of function depends on the duration of RNAi induction. The observed phenotypes might therefore be stronger at the later time point of development. For clonal loss-of-function analyses the FLP/FRT flip-out was employed[62,63]. The *Dfd > Dfd^RNAi* and *Dfd > Dg^RNAi* experiments were performed at 25 °C. In this condition, a weak Dfd staining can be observed (Supplementary Fig. 3a). Increasing the temperature to 29 °C causes early pupal death (or head eversion failure and disc fusion defects) in most of the animals (>50%), precluding analyses of Dfd function in such condition. For clonal loss-of-function analyses larvae were raised at 25 °C, heat-shocked at 37 °C for 12 min (*Tollo^RNAi*) or 45 min (*Dfd^RNAi*) in the third instar larval stage, and then kept at 29 °C for 48 h (*Tollo^RNAi*) or 72 h (*Dfd^RNAi*) before analyses.

### Generation of Dfd-GAL4 and Dg::GFP

The *Dfd-GAL4* and *Dg::GFP* alleles were generated by CRISPR/Cas9-mediated homologous recombination at their endogenous loci, using the vas-Cas9 line[64]. To generate the *Dfd-GAL4* and *Dg::GFP* alleles by CRISPR/Cas9-mediated homologous recombination, guide RNAs were cloned into the *pCFD5:U6:3-t::gRNA* vector[65]. For the *Dfd-GAL4* allele, the following guide RNAs were used: 5′-AGA AAA GAG CTC ATG ACG GAA GG-3′ and 5′-CAT GAG AAA AGA GCT CAT GAC GG-3′; while for the *Dg::GFP* allele, we used the following ones: 5′-TGG TTT GCG GTA GGA GGG CGT GG-3′ and 5′-TGG CGG TGG TTT GCG GTA GGA GG-3′. Homology sequences were cloned into homologous recombination vectors harboring a hs-mini-white cassette flanked by two loxP sites[66,67] and an N-terminal GFP sequence for GFP tagging of Dg or an N-terminal GAL4 sequence to insert GAL4 downstream of the *Dfd* promoter. The vector used for GFP tagging has been described in [66]. The vector used to insert GAL4 was generated by replacing the mKate2 sequence with a

GAL4 sequence in the mKate2 tagging vector[66] using the following primers: 5′-ATG AAG CTA CTG TCT TCT ATC GAA CAA GC-3′ and 5′-ATA GCA TAC ATT ATA CGA AGT TAT GGA TCC TTA CTC TTT TTT TGG GTT TGG TGG GGT ATC-3′. To generate the *Dfd-GAL4* line, the two homologous regions (HR1 and HR2) flanking the site of CRISPR/Cas9 cuts were cloned using the following primers: (HR1) 5′-CCC GGG CTA ATT ATG GGG TGT CGC CCT TCG CGA GGG TAG GTA AGT AGG TGT GC-3′ and 5′-TGC TTG TTC GAT AGA AGA CAG TAG CTT CAT CAT GAC GGA AGG TCT GTT GGA TCG-3′; (HR2) 5′-AGT TCG GGG TCC AGC GGT TCT TCA GGC AGT AGC TCT TTT CTC ATG GGT TAT CCG C-3′ and 5′-ATT TTG TGT CGC CCT TGA ACT CGA TTG ACG CTC TTC GAC TAT GCG GCA TAA CAA TCT TGT GGG C-3′. The following HR1 and HR2 were cloned in the GFP tagging vector for Dg::GFP: (HR1) 5′-CTA ATT ATG GGG TGT CGC CCT TCG GGT CTC TAG TTG AAC GAA GAG TTC TAT GGC ATT CCG-3′ and 5′′-GAA CTG CCT GAA GAA CCG CTG GAC CCC GAA CTG GAG GGC GTG GCT GGC GAC TTG-3′; (HR2) 5′-CTC GGG AAG TGG TAG CTC AGG GTC TAG TGG ATA CCG CAA ACC ACC GCC ATA TG-3′ and 5′-GCC CTT GAA CTC GAT TGA CGC TCT TCG TCC GAA TTA TCC AAA GGG GAG CTT GTG-3′. Cloning was performed using SLIC[68]. All embryo transgenesis injections were performed by Bestgene and transgenes were confirmed by sequencing. Detailed plasmid maps and their DNA sequences are available upon request.

## Pupa and adult mounting and imaging

**Time-lapse live imaging and tissue curvature modulation.** Pupae were collected at 0 hAPF or at head eversion (12 hAPF). Pupa mountings for live-imaging were adapted from[66,69] to reproducibly ensure (i) medial flattening, (ii) lateral flattening, or (iii) no flattening of the dorsal neck tissue. At approximately 14–17 hAPF, the dorsal part of the head and thorax pupal case was removed using dissection Vannas spring scissors and Dumont #5 tweezers (Fine Science Tools). *For medial flattening*: Pupae were then adhered ventrally on a slide using double-sided tape (Scotch 3 M) and the pupae heads were elevated by placing 6 layers of double-sided tape below them. Two spacers respectively made of six and five 18 × 18 mm #1 (0.13–0.16 mm) coverslips (Thermo Scientific, Menzel-Glaser) were placed posteriorly and anteriorly to the pupae, respectively, close to the edges of the slide. The spacers supported the top 24 × 40 mm #1 (0.13–0.16 mm) coverslip (Knittel Glass) placed over the pupae and covered with a fine layer of mineral oil 10 S (VWR International). The top coverslip was glued to the spacers using nail polish. This resulted in a flattening of a tissue region centered on the midline. *For lateral flattening*: the mounting was similar except that the pupae were adhered ventrally slightly sideways to yield a lateral flattening upon gluing the top coverslip on the two spacers. *Without coverslip flattening:* to avoid flattening of the dorsal region upon mounting of the top coverslip, the spacers were made respectively of eight and seven 18 × 18 mm coverslips to prevent direct contact between the coverslip and the pupae. To allow imaging in this case, a thicker layer of mineral oil was spread over the coverslip before placing it on top of the pupae. By gently moving the coverslip when placing it in contact with the two spacers, a contact zone between the mineral oil and the presumptive neck region could be created. Care was taken to restrict immersion into mineral oil to only the most dorsal part of the pupae to prevent the oil from covering the lateral spiracles and disrupting breathing.

Samples were imaged at 25 °C or 29 °C with an inverted confocal spinning disk microscope from Nikon or Carl Zeiss using 40×/1.4 OIL DIC H/N2 PL FLUOR, 60×/1.4 OIL PL APO, 63×/1.4 OIL DICII PL APO or 100×/1.4 OIL DIC N2 PL APO VC objectives and a sCMOS camera (Orca Flash4, Hamamatsu). For comparison of live-imaging quality with and without coverslip, 18 hAPF Ecad::3xGFP expressing pupae were sequentially imaged using a 40×/0.6 DRY CFI S Plan Fluor ELWD objective without oil and subsequently a 40×/1.4 OIL DIC H/N2 PL FLUOR objective with oil under identical imaging parameters.

To record neck invagination, 14–17 hAPF old Ecad::3xGFP or ubi-Ecad::GFP expressing pupae were imaged until approximately 24 hAPF every 15 min, using 60-step z-stacks with 1 μm spacing. Autofocus detection (Metamorph software) was used to keep the focus on the tissue throughout the time of imaging.

**Pupal tissue fixation, staining, and imaging.** Pupae were dissected, fixed, and stained as previously described[70]. Primary antibodies used: mouse anti-Antp (DSHB #8C11, 1:20 dilution), rabbit anti-Dfd (1:100 dilution, gift from T. Kaufman). Secondary antibodies used: donkey anti-rabbit Cy5 (Interchim, Cat.#: 711-175-152, 1:500), donkey anti-mouse Cy3 (Interchim, Cat.#: 711-165-152, 1:500). Phalloidin Abberior STAR RED (Sigma, Cat.#: 30972-20UG, 1:2000) or Phalloidin Alexa Fluor 647 (ThermoFisher, Cat.#: A22287 1:1000) were used to label F-actin. Fixed tissues were mounted in 80% Glycerol (Sigma) in 1×PBS supplemented with 2% N-propyl gallate (Sigma) and imaged using an inverted confocal spinning disk microscope from Nikon or Carl Zeiss using either 40×/1.4 OIL DIC H/N2 PL FLUOR, 60×/1.4 OIL PL APO, 63×/1.4 OIL DICII PL APO or 100×/1.4 OIL DIC N2 PL APO VC objectives and sCMOS camera (Orca Flash4, Hamamatsu).

**Ex vivo imaging of the basal surface of the pupal dorsal epidermis.** To better visualize the basal surface of the neck epidermis, an ex vivo live or fixed imaging approach was adapted from[71]. Upon removal of the pupal case, pupae were glued on their dorsal side to a 20 mm glass bottom microwell dish (MatTek Life Sciences) using heptane glue[72]. They were then dissected in 1×PBS as previously described[73] and imaged using an inverted confocal spinning disk microscope from Nikon or Carl Zeiss using either 40x/1.4 OIL DIC H/N2 PL FLUOR, 60x/1.4 OIL PL APO or 100x/1.4 OIL DIC N2 PL APO VC objectives and sCMOS camera (Orca Flash4, Hamamatsu). In case of imaging fixed samples, dissected tissues were fixed and stained as described above.

**Injections.** A 22 h hAPF pupae expressing Ecad::3xGFP were glued on their dorsal side to a 20 mm glass bottom microwell dish (MatTek Life Sciences) using heptane glue[72] and imaged at 10 min intervals (80 z-stacks with 1 μm spacing) for 1.5–2 h using an inverted confocal spinning disk microscope from Carl Zeiss using a 40×/1.4 OIL DIC H/N2 PL FLUOR objective. After having recorded the pre-injection neck invagination dynamics, the dish was removed from the microscope, and pupae were manually injected under a stereo microscope (Carl Zeiss), using a NanojectII system (Drummond Scientific), either with 27.6 nL H$_2$O or 50 mM Y-27632 (Tocris) dissolved in H$_2$O in the abdomen. Upon injection, pupae were imaged for an additional 1.5–2 h to record the post-injection invagination dynamics.

**Whole animal and head imaging.** Whole adult flies and pupae were imaged in glycerol:ethanol (4:1) using a Carl Zeiss Stereo Discovery V20 fluorescent microscope equipped with an Axiocam camera using a PlanApoS 1.0x FWD 60 mm objective. Upon separating the adult head from the rest of the body by cutting the adult neck, adult head pictures (Supplementary Fig. 1e) were obtained by gluing the anterior part of the head on a slide and by imaging the cuticular autofluorescence of the back of the head with a spinning disk confocal microscope using a 10×/0,3 DIC L/N1 PL FLUOR objective and 488 nm excitation laser.

## Laser ablations

Laser ablation can be used as a probe, to measure the existing tension, usually by severing the apical, lateral, or basal cortex domain of the cells: tissue recoil velocity immediately after ablation is considered to be a reliable proxy of the tension which existed in the tissue immediately before ablation[33,42]. We performed such probe ablation using a multi-photon laser. Laser ablation can also be used to sever the tissue across its whole thickness: it artificially creates a physical opening within the tissue, thus mechanically isolating a piece of tissue from

another piece and providing a stress-free boundary. We performed such ablations using a UV 355 nm laser.

**Probe laser ablation and quantification of recoil velocity.** To measure the apical and basal tissue recoil velocities after ablation, laser ablations were performed using an inverted laser scanning microscope (LSM880 NLO, Carl Zeiss) equipped with a multi-photon Ti::Sapphire laser (Mai Tai HP DeepSee, Spectra-Physics). Ablations were performed in 14–24 hAPF pupae expressing the utrABD::GFP actin or Ecad::3xGFP AJ markers mounted with medial coverslip flattening and imaged in single-photon bidirectional scan mode every 1 s for 30 s with a 40×/1.3 OIL DICII PL APO (UV) VIS-IR (420762-9800) objective. A region of interest (ROI) corresponding to a $34.5 \times 17.25\,\mu m$ rectangle along the AP axis and centered on the medial or lateral neck region was ablated as previously described[69,74,75]. Apical and basal ablations were sequentially performed in the same pupa. The initial tissue recoil velocity, which is proportional to the stress[33] was determined between $t = 1\,s$ and $t = 7\,s$ after ablation. No differences in the recoil velocities were found upon changing the order of sequential ablations of the apical and basal domains (Supplementary Fig. 6i, j). To measure AP and ML tension simultaneously, either basally or apically, a small $8.6 \times 8.6\,\mu m$ circular ROI was ablated in 22 hAPF-old pupae and imaged every 0.25 s. The initial recoil velocity was measured between 0.25 s and 1.25 s following ablation. Note that recoil velocities after ablation indicate tensions only up to a prefactor (which is typically a dissipation factor). They can be considered as reasonable proxies of tensions[33,42], while absolute tension values are not measured.

The same microscope set-up was used to measure recoil velocity along the apical-basal axis after lateral ablation. Ablations were performed in 21–22 hAPF Ecad::3xGFP pupae. Before ablation, a $21 \times 21\,\mu m$ stack (50–60 z-planes, 0.5 μm apart) centered on the boundary between the thorax and neck was acquired to visualize the apical AJ Ecad::GFP signal and the weaker basal Ecad::GFP signal. Ablation was then performed in a $21 \times 21\,\mu m$ ROI located at a z-plan equidistant from apical and basal tissue surfaces. The $21 \times 21\,\mu m$ stack (50–60 z-planes, 0.5 μm apart) was subsequently acquired 30 s after ablation. The lateral membrane recoil velocity was determined by measuring the distance between apical and basal tissue domains before and 30 s after ablation. Since this laser power is sufficient to ablate the actomyosin network of the tissue at a deeper position to trigger tissue recoil, we can safely consider that this regime is sufficient to ablate the lateral cortex to estimate recoil velocity.

**Tissue severing laser ablation to mechanically isolate neck tissue.** To mechanically isolate the presumptive neck tissue from head and thorax tissues, ablations were performed on an inverted spinning disk wide homogenizer confocal microscope CSU-W1 (Roper/Carl Zeiss) equipped with a UV 355 nm laser ablation module and an sCMOS camera (Orca Flash4, Hamamatsu). Prior to neck deepening (e.g., prior to 17 hAPF) or at 21 hAPF, a head and a thorax tissue ROI of ~330 × 50 μm flanking the medial dorsal presumptive neck region were ablated. Ablations were performed using a UV 355 nm laser with a power of 0.1 mW using the 40×/1.4 OIL DIC H/N2 PL FLUOR objective and across 60 imaged z-planes (1 μm spacing). Ablations were repeated every 3 h. To mechanically isolate the medial presumptive neck tissue from lateral neck tissues, a similar set-up was used to ablate a left and a right ROI (~50 × 330 μm) flanking the medial neck tissue. Using this ablation regime, we verified that we ablate the tissue across its whole thickness (Supplementary Fig. 11f).

### Quantifications of tissue dynamics

**Spatial and temporal registration of time-lapse movies.** Between 14–17 hAPF the presumptive neck-thorax boundary, the most medial and posterior pair of head macrochaetae, the anterior macrochaetae of the scutum, and the midline provide reproducible spatial landmarks easily identifiable using cortical markers such as Ecad::3xGFP (Supplementary Fig. 1c). We used their positions before 18 hAPF to register all time-lapse movies in space. Along the ML x-axis, the midline position was defined as $Position_{ML} = 0\%$, and the most posterior and medial pair of head macrochaetae was set at $Position_{ML} = -100\%$ (left macrochaeta) and $Position_{ML} = 100\%$ (right macrochaeta), respectively (Supplementary Fig. 1c). Along the AP y-axis, the initial position of the presumptive neck–thorax boundary in a ML box at $Position_{ML} = -50\%$ to 50% was defined as $Position_{AP} = 0\%$, and the average AP position of the anterior scutum macrochaetae was defined as $Position_{AP} = 100\%$. In all time-lapse movies, the deviation angle of the AP axis from the vertical axis in the xy plane, as well as the deviation angle of the AP axis from the horizontal axis in the xz plane did not exceed 15° and were corrected during data analysis.

To register time-lapse movies in time, we measured in each movie the time at which all microchaete precursors located between $Position_{AP} = 20\%$ and 50% underwent their last division and set this time at 22 hAPF. This last division measurement provided a timing with a 30 min precision ($N = 11$ control animals). To correct for the difference in developmental time between 25 and 29 °C, a correction factor of 1.27 was applied to data acquired at 29 °C to match the 25 °C timing[76]. Note that each experimental condition was compared to a control condition acquired at the same temperature, so this correction factor does not impact any of our conclusions.

**Tracking of apical and basal neck fold front.** The apical neck fold front was tracked in 3D in animals expressing Ecad::3xGFP or ubi-Ecad::GFP from confocal time-lapse movies along the ML x-axis, the AP y-axis, and the AB z-axis. The boundary between the presumptive neck cells and the thorax cells (coinciding with the apical fold front) was first tracked in time on z-maximal projections. At each time point, an ROI of 10– 20 μm centered along the boundary (Supplementary Fig. 1c) was used to generate a transverse maximal projection in the ML and apical-basal plane (Supplementary Fig. 1d). Any global imaging drift during acquisition was then corrected using fixed particle dust on the apical ECM using the transverse view. Note that the boundary is roughly straight on z-projections, the curvature of the boundary in the ML and AP plane was therefore not taken into consideration. On each transverse section and using either the Ecad::3xGFP or ubi-Ecad::GFP AJ signal, the apical neck fold front was manually segmented as a line by placing a landmark approximately every 20 μm. For quantification, each manually segmented fold front position was then interpolated along its length with regular sampling spacing using the *interparc* function of Matlab (https://www.mathworks.com/matlabcentral/fileexchange/34874-interparc). The basal neck fold front was tracked using a similar approach using the most basal Ecad::3xGFP or ubi-Ecad::GFP signal. Tissue thickness was defined as the distance between the Ecad::3xGFP AJ and basal signals in transverse sections.

**Neck fold deepening.** Neck fold deepening was measured by tracking in time of the apical fold front. $Position_{ML}$ was tracked in time by considering that each $Position_{ML}$ along the front linearly moves in time towards a center of convergence (Supplementary Fig. 1e, f). The center position of convergence was estimated as being at the midline $Position_{ML} = 0\%$ and the AB position corresponding to the final position of the neck in a posterior view of an adult head, taking into consideration the impact of flattening due to the mounting procedure (Supplementary Fig. 1e, f). To determine neck fold deepening, for each time point $t$ and each $Position_{ML}$, neck fold depth was estimated by calculating the distance between the initial position of the neck fold front at 16 hAPF (or at the onset of imaging for the injection experiments) and the corresponding neck fold front $Position_{ML}$ at subsequent time points. For each animal, neck fold depth was then averaged between $Position_{ML} = -250\%$ and $+250\%$.

**Fold front curvature and local normal deepening speed.** For each time point $t$ and at given $Position_{ML}$ (centered on $Position_{ML} = 0\%$ and evenly spaced in 10% increments on the tracked apical fold front, Supplementary Fig. 1g, h), local apical neck fold curvature was estimated by fitting a circle on a triplet of points along the tracked apical fold front corresponding to the ML point at which curvature is estimated and the two points located 22.5 μm away on either side of this ML point. Local curvature was defined as the inverse of the radius of the fitted circle. Using the same triplet of points, the local normal deepening speed was calculated as follows. For each time point $t$ and each $Position_{ML}$, the normal deepening speed was calculated by measuring the distance between the considered $Position_{ML}$ at time $t$ and the intersection between local normal and the fold front $Position_{ML}$ at time $t + 1$. Normal deepening speed and curvature were smoothed in time using a 1 h 45 min sliding window. For each time point $t$ and each $Position_{ML}$, the average and the standard error of the mean (sem) of front curvature and deepening speed were calculated among animals of a given condition. A few ML (less than 2% of the total) positions located on the edges of the tracked fold front were sampled in less than 5 animals (usually corresponding to $Position_{ML} < -300\%$ or $Position_{ML} > 300\%$ at the beginning of the time-lapse movies) or showed a curvature characterized by a sem higher than 0.002: they were not taken into consideration in subsequent analyses.

**Transverse section kymograph.** Transverse projections of the neck fold front were used to generate a kymograph by plotting at each time point a 4 μm wide region of the tissue centered around $Position_{ML} = 140\%$.

**Tissue velocity measurements.** Local tissue velocities were measured either manually by tracking cell displacements over time in Fiji (see an example in Fig. 1b) or by using particle image velocimetry (PIV) on $z$-maximal projections of the Ecad::3xGFP AJ signal using previously published custom Matlab codes (see an example in Supplementary Fig. 6n)[69].

**Detection of the apical surface for apical or basal projection.** To analyze protein distributions on the apical or basal surface of the neck fold on a confocal $z$-stack, fluorescent signals were projected using a combination of custom Fiji scripts and Matlab codes that automatically determines the apical $z$-map using an apical marker such as the AJ Ecad, F-actin, or MyoII markers. The confocal $z$-stack was first preprocessed in Fiji to detect the most apical position of high variance for each pixel as follows: (i) each $z$-section was filtered using Fiji Subtract Background plugin (rolling ball radius of $2 \times 0.322$ μm), the resulting $z$-stack was then downscaled by a factor of 10 and a variance filter was applied along the $z$ direction; (ii) using the *peakseek* Matlab function (https://www.mathworks.com/matlabcentral/fileexchange/34874-interparc), the $z$-position of each pixel was detected to define the most apical position of high variance for each pixel; thus generating a topographic map of the apical surface. The topographic map was then up scaled to the original image size to produce the apical $z$-map of the tissue. This apical $z$-map can then be used to project any signal at any apical or basal position.

**Cell shape measurements.** Measurements of apical and basal AP and ML cell length were performed on PH::GFP $z$-stack images (150 slices at 0.5 μm spacing and 0.065 μm $xy$ resolution) captured at 16 and 24 hAPF in Fiji by measuring the longest basal and apical cell widths on sagittal and transverse slices centered around the midline and neck regions (~8 cells wide), respectively. A region of the head-neck-thorax tissues at 16 and 24 hAPF was segmented in 3D by manually tracing the cell outlines in each $z$-slice in Fiji and processed using custom Fiji and Matlab codes. To characterize head, neck, and thorax apical cell shape anisotropies, randomly selected cells within the medial neck region (within ~100 μm

from the midline) were manually segmented on projected $z$-stack images of Ecad::3xGFP at 13, 14.5, 16, 18, and 20 hAPF. The aspect ratio of traced apical cell shapes was measured using Fiji.

**Cell delamination event quantification.** Within a cell patch ($Position_{ML} = -150\%$ to 150% and $Position_{AP} = -25\%$ to 25%) delamination events were manually counted between 16 hAPF and 22 hAPF.

## Quantification of F-actin basal organization

The organization of the F-actin network was quantified on dissected and fixed tissue at 21 hAPF and stained with phalloidin. Both apical and basal F-actin signals were projected separately (see *Detection of the apical surface for apical or basal projection)*. The basal F-actin signal intensity was then normalized by the average apical one. The basal F-actin fibers were automatically segmented by (i) applying the Subtract Background Fiji plugin (rolling ball radius $3 \times 0.161$ μm) and (ii) skeletonizing them using a threshold value allowing to segment most of them. The segmented F-actin fibers were then processed using the Fiji Analyze Skeleton plugin to remove F-actin fiber "vertices" and keep a linear portion of each F-actin basal fiber. To ensure a meaningful measurement of F-actin fiber orientation, the orientation of the resulting segmented F-actin basal fibers of length larger than 0.8 μm was measured with respect to the ML axis (0° and 90° being, respectively, parallel and perpendicular to the ML axis). The length of each fiber was estimated by the distance between its two edges. Average basal F-actin fiber orientation was quantified as a weighted average of the F-actin fiber orientation and length in different ROIs: a $57 \times 123$ μm ROI centered at the presumptive neck-thorax boundary, a $23.5 \times 123$ μm ROI located at the neck–thorax boundary on the thorax side or a $23.5 \times 123$ μm ROI located at the neck–thorax boundary on the head side.

## Quantification of Myosin levels as a function of neck curvature

Neck folding was recorded in pupae expressing both Ecad::3xGFP and MyoII::3xmKate2. Curvature along the ML axis was estimated in 21 hAPF-old pupae in the same images using the method described in *Fold front curvature and local normal deepening speed*. The intensity of MyoII::3xmKate2 and Ecad::3xGFP signals were estimated for each time point in top projections centered on the apical plane (see *Detection of the apical surface for apical or basal projection*). First, the background was removed by applying the Subtract Background Fiji plugin (rolling ball radius $200 \times 0.322$ μm), and signals were smoothed in space using a median filter (size $20 \times 0.322$ μm). MyoII::3xmKate2 and Ecad::3xGFP levels were then measured along the ML axis by isolating a line manually drawn in the middle of the neck region. In each animal, the MyoII::3xmKate2 and Ecad::GFP signals along the ML axis were calculated and binned according to curvature (0.001 μm$^{-1}$ bins), to calculate an average value for MyoII::3xmKate2 and Ecad::GFP signals for each curvature bin. For each time point, the MyoII::3xmKate and Ecad::GFP values of each curvature bin were normalized by the average value for all curvatures. ANOVA test was used to determine whether signal intensity varied across the ML axis.

## Image processing for display

Unless otherwise stated, all images and movies for display represent maximal $z$-projections and were subjected to image processing (denoising and contrast enhancement) using Fiji[77] or Matlab. 3D representations were obtained using either the Imaris software (https://imaris.oxinst.com/), the Interactive 3D Surface Plot plugin in Fiji, or Matlab.

## Statistics and reproducibility

Sample sizes vary in each experiment and animal samples were randomly selected within a given genotype for subsequent analyses. Experiments were repeated at least twice. Only animals correctly

mounted for microscopy and without developmental delay were included in the analyses. The numbers of analyzed animals, ablations, and analyzed cells are indicated in figure legends. For all graphs, each error bar/error area represents the standard error to the mean (sem), except for ablation, apical cell aspect ratio, basal fiber orientation, and delamination for which standard deviation (sd) is shown. For all figures, representative microscopy images of at least two different experiments are shown. In addition, upon averaging among animals of the same genotype, the sd associated with the slopes of linear fits with a zero intercept were extracted from datasets of deepening speed as a function of the product of curvature ($\kappa$) and initial recoil velocity estimated by laser ablation ($v$). The corresponding errors are $\sigma_\kappa$ and $\sigma_v$, respectively, and the error $\sigma_{\kappa,v}$ associated with their product was estimated as: $\sigma_{\kappa,v} = \bar{\kappa}\sigma_v + \bar{v}\sigma_\kappa + \sigma_\kappa\sigma_v$ with, respectively, $\bar{\kappa}$ and $\bar{v}$ the average curvature and velocity values among animals. Experimental data used for recoil velocity, curvature, and deepening speed are given in figure legends. The statistical tests used to assess significance are stated in the figure legends and are two-sided. For comparing deepening dynamics (or the evolution over time of the initial recoil velocity after ablation) between two conditions, a Welch test was performed for each time point, and the associated $p$-value was plotted as a horizontal bar (white: $p > 0.05$, striped: $p < 0.05$ and plain: $p < 0.01$) on top of each graph. Exact $p$ values are provided in the Source data file. Comparisons of two distributions (initial recoil velocities, number of delamination events, or orientation of basal fibers) were performed using Welch tests. $t$-tests were performed to analyze whether average changes in the cell area, cell AP length, and cell ML length were different from 0. ANOVA was used to evaluate the significant differences of MyoII::3xmKate2 or Ecad::3xGFP signal intensities as a function of curvature, and to test whether the apical cell aspect ratio of head, neck, and thorax cells at 13 hAPF was different. Except for ANOVA, no multiple comparison adjustments were made. Statistical analyses were performed using Matlab, Excel, and GraphPad Prism.

### Reporting summary

Further information on research design is available in the Nature Portfolio Reporting Summary linked to this article.

## Data availability

The image datasets generated are available from the corresponding author upon reasonable request. Source data used to generate each plot are provided in this paper. Source data are provided in this paper.

## Code availability

Scripts and codes are accessible at https://doi.org/10.5281/zenodo.7521478.

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

## Acknowledgements

We thank Y. Hong, L. Johnston, R. Karess, T. Kaufman, T. Lecuit, V. Mirouse, J. Zallen, the Bloomington, Vienna, Harvard Medical School Stock Centers, and the Developmental Studies Hybridoma Bank for reagents; the PICT-IBiSA@BDD imaging facility (ANR-10-INBS-04); A. Bardin, V. Cachoux, F. Gallet, E. Hannezo, P. Léopold, J.-L. Maître, F. di Pietro, D. Pinheiro for comments, O. Renaud, O. Leroy for help with ablation, A. Dauphin for laser power measurements, A. Maugarny-Calès for advice on injections, C. Roffay, S. Rigaud, and V. Cachoux for inputs regarding image analyses or visualization. This work was supported by Institut Curie, CNRS, INSERM, ANR-Migrafolds, ERC Advanced (340784), ARC (SL220130607097), ANR Labex DEEP (11-LBX-0044, ANR-10-IDEX-0001-02). A.V. and L.A. acknowledge FRM (FDT201805005805) and ARC (PDF20181208399) fellowships, respectively.

## Author contributions

A.V., L.A., F.G., F.B., and Y.B. designed the project. I.G. and A.J. produced reagents. A.V., L.A., and F.B. developed experimental methods and performed experiments. A.V., F.G. developed data analysis and modeling. A.V. developed quantification scripts. A.V., L.A., and F.B. analyzed the data. A.V., L.A., F.G., F.B., and Y.B. wrote the paper.

## Competing interests

The authors declare no competing interests.
