## [Peer Review File · Nature Communications]

Homeotic compartment curvature and tension control
spatiotemporal folding dynamicsREVIEWER COMMENTS

Reviewer #1 (Remarks to the Author):

Both biochemical factors and mechanical forces have been shown to play important role in regulating tissue shape change during morphogenesis. However, an interesting yet unanswered question is how original shape of the tissue influence its response to mechanical or biochemical cues. In this work, Villedieu and colleagues present evidence that during *Drosophila* neck fold formation, the initial tissue curvature, together with the in-plane tension generated by myosin II contractility at both apical and basal side of the neck cells, determines how tissue folding proceeds. The authors show that the generation of in-plane tension is controlled by the Deformed (Dfd) homeotic gene through the Tollo and Dystroglycan (Dg) receptors. Through elegant quantitative image analysis, the authors show that the rate of neck fold deepening is largely proportional to the product of in-plane tissue tension and local tissue curvature under various conditions tested, including genetic manipulation of actomyosin contractility and alteration of tissue curvature by surface flattening or laser disruption of lateral regions of the neck domain.

The mechanism proposed in this study is distinct from the mechanisms previously identified in other tissue folding processes and would be of immense interest to the field. The experiments are well designed, and the analysis and presentation of the data are overall of very high quality. However, given the complexity of the tissue context and potential pleiotropic effect of certain treatment used in this study, further clarifications are required for some of the results and conclusions. In addition, while the evidence that supports the role of tissue curvature in neck folding is compelling, the lack of a clear picture about the biochemical regulations of apical and basal tension is a limitation of the study. The detailed comments are listed below.

Major comments:

1. The authors propose that the generation of medial-lateral tension in the neck region is controlled by Tollo and Dg/Dys downstream of Dfd. However, disruption of the function of these genes by RNAi or by genetic mutations only generated moderate defects in neck folding (Fig. 2a and 2g). It is unclear whether the lack of stronger defects is due to a partial disruption of the target gene or the presence of other parallel pathways that facilitate the generation of medial-lateral tension. If the former is the case, would a stronger disruption of the target genes completely prevent neck folding (e.g., using null alleles if feasible)? The authors should assess the effect of RNAi (e.g. by immunostaining or fluorescently tagged proteins) and also comment on the nature of the genetic mutants used in this study. In addition, to help further understand the link between the upstream regulators and tension generation, the temporal profile of Dfd, Tollo and Dg expression in the neck region, in particular the earliest stage when they can be detected, should be described. Finally, it would be helpful to indicate the lateral border of Dfd expression in the neck domain and check whether it overlaps with the border of the high medial-lateral tension region of the neck domain.

2. The authors' model argues for a critical role for actomyosin-dependent tissue tension in neck folding; however, disrupting myosin II or its activator Rok by RNAi only partially inhibited the folding process (Fig. 1k). While this could be explained by a partial knockdown effect, the lack of a stronger inhibition of actomyosin prevents the test of whether a complete loss of tension during the 16 – 24 hAPF window would block neck folding all together. The current data therefore could not entirely rule out the potential contribution of additional mechanism(s) that operates at later phase of the folding process. Is there a way to completely abolish actomyosin contractility/in-plane tension prior to or during neck folding (e.g., treatment with Rok inhibitor or actin depolymerization drug)? Alternatively, a test of the immediate impact of disrupting tissue tension may be achieved by laser ablation at the medial region of the neck domain during the folding process, which should result in tissue recoil that substantially reduces tissue tension.

3. The way how Tollo and Dg/Dys activate actomyosin in the neck folding context is unclear. While delineating the entire pathway appears to be beyond the scope of this paper, the observation that disruption of Tollo (apical localized) or Dg/Dys (basally localized) affected both apical and basal tension to a similar extent is quite unexpected and worth additional investigation. As summarized in Extended Data Fig.4l, the authors propose the presence of feedback between apical and basal tensions. However, an alternative explanation is that Tollo and Dg/Dys are mutually dependent for their appropriate localization (and/or function). This possibility should be tested.

4. The tissue ablation experiment presented in Figure 4i-4n is an important experiment designed to test the role of tissue curvature in neck folding. The phenotype is striking. However, at the current form, it is difficult to determine whether the observed effect on neck folding is due to direct alteration of curvature or due to some indirect effects. First, it is unclear at which stage the ablation was performed. If the neck region is already under medial-lateral tension at the time of ablation, removing the lateral sides of the neck region would be expected to cause recoil of the remaining neck tissue towards the midline, resulting in a reduction of medial-lateral tension. If this happens, it may complicate the interpretation of the results. Second, considering potential secondary effect of laser ablation (e.g., generation of heat) on tissues neighboring to the ablated region, the authors should examine whether the treatment affects myosin/actin distribution and/or actomyosin-dependent tissue tension in the medial region of the neck. Finally, evidence that the laser treatment physically isolated the medial neck region should be demonstrated.

5. Regarding the analysis designed to rule out the contribution of apical constriction/cell wedging (Extended Data Fig.5a – c), it is unclear why the authors select 20 hAPF and 22 hAPF for comparison. It would be more informative to compare the cell morphology between timepoints near the beginning and end of neck folding (e.g. 16 and 24 hAPF, respectively). The selection of stages could be important since in some other folding processes (e.g. Drosophila ventral furrow formation, ascidian endoderm invagination), cell wedging/basal expansion does not happen until later phase of folding. In addition, 3D cell shape should be shown to make the conclusion more convincing.

6. The physical model presented in this study considers a line under tension (Fig. 3c), however, the neck epithelium has a thickness that is non-negligible compared to the size of the neck fold. How tissue thickness may influence the comparison between the real tissue and the computer model should be discussed.

Minor points:

1. In Figure 1b, the authors show that the neck cells are already elongated medial-laterally at 18 hAPF, whereas the cells in the head/thorax regions are more isotropic. What causes this morphological difference between the neck and head/thorax regions? When does this difference first appear? Is Dfd (or Tollo, Dg/Dys) involved in generating the elongated morphology of the neck cells?

2. In figure 1g, h, only 18 hAPF is shown for apical signals, and 21 hAPF is shown for basal signals. To understand how apical and basal tissue tension evolve over time, it would be helpful to show how the apical and basal actomyosin structures change throughout the folding process.

3. The basal tissue tension is non-zero between 14 – 18 hAPF. This observation should be discussed in the light of the finding that “MyoII and F-actin fibers ... being strongly aligned with the ML axis from 21 hAPF onwards”.

4. Extended figure 2: the title “measurements of apical-basal tension” is misleading (it seems to indicate tension along the apical-basal axis). Better restated as “measurements of apical and basal tension”.

5. Extended figure 2i: The red line is missing from this panel. If the apical side moved out of the view

at 24 hAPF, it should be stated in the figure legend.

6. Extended figure 3b: The difference between the mutant clone (outside of the dotted circle) and the control region (within the dotted circle) in the Dfd RNAi sample is very hard to appreciate. If the intention is to compare between the Dfd RNAi sample and the control sample, it should be made clearer in the figure legend.

7. Extended figure 3b and 3c: The basal F-actin signal in the control in panel b appears very different from the basal F-actin signal in the Dfd-GAL4 control in panel c.

8. Extended figure 4k: what is the bright signal at the basal region of the cross-section view? (It is not present in panels 4f – 4j).

9. Extended figure 5f: It is unclear whether ablation occurred or the tissue was simply photobleached. The conclusion would be more convincing if the authors could show evidence that the lateral cell cortex was indeed ablated.

10. Extended Figure 6: The purpose of this figure is to show the myosin distribution; however, only images with Ecad-GFP signal were shown. Myosin images should also be displayed in addition to the quantification plot.

11. Extended Figure 9a: it is interesting that Rok RNAi and Myosin II RNAi mainly influence the basal tension after 18 hAPF, but have no effect on the initial tension increase between 14 – 18 hrs.

Reviewer #2 (Remarks to the Author):

The authors described for the first time the formation of neck folding between head and thorax in *Drosophila* pupae and revealed a new mechanism of fold formation including contribution of acto-myosin-dependent in plane tension (both in basal and apical sides) and medio-lateral (M-L) curvature of the tissue. They combine live imaging and laser ablation in different genetic backgrounds to describe the contribution of in plane tension and curvature of the head-thorax neck fold formation during pupal development. The paper is organized in two parts: First, they identify the homeotic gene *Dfd* as a key regulator of both apical and basal in plane tension through respectively *tollo* receptor expression and *Dg* and *Dys* localization in the presumptive neck compartment. Then, they found a correlation between curvature along M-L axis and deepening speed and addressed the contribution of curvature in tissue depth invagination and deepening speed.

Overall, I found that the results described here are very interesting and of high quality. They should certainly be published in *Nature Communication*. However, I have a number of concerns that I think the authors must address before publication.

Experimental part:

Fig1 Major point: Splitting of current Fig1 in two figures would allow for a more complete description of the tissue's properties. The first as a complete description of the *Drosophila* neck folding during pupal development: since this invagination has never been described, it is necessary to show cell shape, *Dfd* expression and Myosin accumulation at the initial point just prior fold formation (14H APF global view and zoom as in Fig1 2b). If *Dfd* regulates apical and basal tension in these cells, it would be interesting to visualize *Dfd* expression and DE-Cad expression (cell shape) from the onset of the process (14H) and up to 24H. It would be also interesting to describe the mechanical properties of the *Dfd* expressing compartment in comparison with head and thorax tissues (in sup data). Fig1d might be in sup data.

The title of the second figure could be: *Drosophila* neck folding required Myosin- dependent apical and

basal in plane tension. Apical and basal MyoII accumulation should be described for 3 time points (14H, 18H and 21H for example). As mentioned above, showing the initial state (14H) would be informative.

Major points: Concerning ablations reported in Fig1 I and j, it is not clear in which part of the tissue ablation is done (lateral, medial or pool of different locations?). Apical and basal tension must be presented in central versus medial in the neck regions, as currently done in extended data fig6f and g. What about contribution of the head-thorax apical myosin cable? Is it cut in the apical ablation experiments? There is a difference in basal and apical tension, especially at 14H. Could it be due the ablation (or non-ablation) of the apical myosin cable? This difference in tension between apical and basal should be discussed. (discussion)

Minor point: In extended fig1 I wonder how do you define the neck region in panel a. Head cells next to the neck region appears elongated along the M-L axis. Also, it should be informative to delimit in the sup Fig1c panel the neck region, so readers have a notion of the size of the neck relative to the head.

Fig2: Apical Tollo accumulation is clear and down regulation of Myo in RNAi Tollo clone as well. However, Dg:GFP basal accumulation within neck region is not clear (Fig 2e and extended data fig4b). Dg must be presented in a more global view, together with Dfd expression, to better visualize Dg fibers within Dfd expressing cells.

If possible, it would be better to downregulate Dg or Dys within the Dfd expressing cell (using RNAi driven by DfdGal4) and not in the whole tissue since their expression is not specific of the neck region (extended data fig 4b and c).

In Fig 2j actin intensity must be commented, in addition to actin fiber orientation.

When Dg or Dys are downregulated, very few tension is observed in basal as in apical but invagination still occurs (fig 2 g,h i). This result must be commented. May be this could be due to the tissue flow shown in extended data fig5. (discussion)

Extended data Fig3: minor point

Results in a is clear but it is not the case in b. Panel b must be deleted. Actin accumulation is already shown in c (Dfd>DfdRNAi). It would be necessary to show Dg:GFP accumulation depending upon Dfd, but it could be easier to show it in DfdGal4> uasDfdRNAi.

Extended data Fig4: Minor point: The model in L must be presented in Fig2. The data presented in apical show clearly that Dfd might regulate Tollo probably directly but note that in basal Dfd might regulate Dg and Dys localization.

Extended fig5: Major point: In j, ablation at 17H APF reduces depth invagination indicating that flow might contribute to initiation of the process (reduction of depth is comparable to the one observed with Dg or Dys mutants Fig 2g). Please discuss this point in the discussion.

Fig4: Major points: In d the recoil velocity values used from Fig1i are measured in a flattened tissue, although the tissue is not flattened this time. Please use recoil velocity measured in the non-flattened tissue to present your graph. These measures could be presented in extended data.

The curvature must be presented in non-flattened tissue in c.

In h, I wonder what is the best recoil velocity data set to use here from flattened (currently presented in Fig 1i) or non-flattened tissue (see above)?

Minor point: The result with lateral flattening must be more exploited in terms of central versus lateral tension. In i comment the fact that in the neck region cells are still elongated, suggesting that they are still under ML tension, whereas invagination does not occur.

Extended data fig10: minor point: d panel might be transferred in figure 4.

Major point: Discussion should be developed, including some of the points mentioned previously. Interplay between apical and basal tension is very interesting. The authors could propose an

hypothesis to account for the interplay between apical and basal tension.

The flattening conditions might induce compression on the tissue. Therefore, all the experiments that modulate curvature could also alter concomitantly at least another property of the tissue. It must be discussed at the end of the ms.

Minor point: Finally, the neck region is frequently called homeotic compartment, as in the title of the ms or in the last chapter. However, homeotic regulation is not the main point of the ms, so authors could remove mention to this when not clearly tested (as is the case in the last chapter).

REVIEWER COMMENTS

Reviewer #1 (Remarks to the Author):

Both biochemical factors and mechanical forces have been shown to play important role in regulating tissue shape change during morphogenesis. However, an interesting yet unanswered question is how original shape of the tissue influence its response to mechanical or biochemical cues. In this work, Villedieu and colleagues present evidence that during *Drosophila* neck fold formation, the initial tissue curvature, together with the in-plane tension generated by myosin II contractility at both apical and basal side of the neck cells, determines how tissue folding proceeds. The authors show that the generation of in-plane tension is controlled by the Deformed (*Dfd*) homeotic gene through the Tollo and Dystroglycan (*Dg*) receptors. Through elegant quantitative image analysis, the authors show that the rate of neck fold deepening is largely proportional to the product of in-plane tissue tension and local tissue curvature under various conditions tested, including genetic manipulation of actomyosin contractility and alteration of tissue curvature by surface flattening or laser disruption of lateral regions of the neck domain.

The mechanism proposed in this study is distinct from the mechanisms previously identified in other tissue folding processes and would be of immense interest to the field. The experiments are well designed, and the analysis and presentation of the data are overall of very high quality. However, given the complexity of the tissue context and potential pleotropic effect of certain treatment used in this study, further clarifications are required for some of the results and conclusions. In addition, while the evidence that supports the role of tissue curvature in neck folding is compelling, the lack of a clear picture about the biochemical regulations of apical and basal tension is a limitation of the study. The detailed comments are listed below.

We thank the reviewer for her/his positive comments on our experimental work and the importance of our findings. We have addressed all her/his constructive points to improve our manuscript as detailed below.

Major comments:

1. The authors propose that the generation of medial-lateral tension in the neck region is controlled by Tollo and *Dg/Dys* downstream of *Dfd*. However, disruption of the function of these genes by RNAi or by genetic mutations only generated moderate defects in neck folding (Fig. 2a and 2g). It is unclear whether the lack of stronger defects is due to a partial disruption of the target gene or the presence of other parallel pathways that facilitate the generation of medial-lateral tension. If the former is the case, would a stronger disruption of the target genes completely prevent neck folding (e.g., using null alleles if feasible)? The authors should assess the effect of RNAi (e.g. by immunostaining or fluorescently tagged proteins) and also comment on the nature of the genetic mutants used in this study.

We now clearly indicate in the manuscript that we used genetic null conditions for Tollo, *Dys* and *Dg*. In the case of *Dfd*, we cannot use a null allele since *Dfd* is embryonic lethal. We have therefore checked in the *Dfd*>*Dfd*^{RNAi} condition (at 25°C) the level of *Dfd* protein in the neck. We found that *Dfd* staining is almost completely absent in this experimental condition (Extended Data Fig. 3a*¹). Therefore, this

¹ Fig. or Extended Data numbers with an asterisk refer to the revised manuscript. The text modifications addressing the reviewers' comments are in green in the revised version of the manuscript.

experimental condition is strong, but not a null condition. Increasing the temperature to 29°C increases Gal4 activity, and hence Dfd knock-down, results in early pupal death (or head eversion failure and disc fusion defects) in most of the animals (>50%), precluding analyses of Dfd function in this condition. We now state the nature of the alleles used in the text: “a result that was further confirmed by analyzing *Tollo* null mutant animals (Fig. 3a and Extended Data Fig. 4d,e)” and “...we found that loss of Dg or *Dys* function (using *Dg* and *Dys* null mutant animals) as well as inhibiting Dg function in the neck (*Dfd>Dg^{RNAi}*) also slowed down folding (Fig. 3h and Extended Data Fig. 4d,f)”. We now also indicate that we use a strong but not null condition for Dfd: “Strongly reducing Dfd function in the neck (*Dfd>Dfd^{RNAi}*) lowered fold deepening while diminishing apical and basal tensions (Fig. 3a-c and Extended Data Fig. 3a,c)”. Last, in Methods (page 21, line 10), we now indicate that the *Dfd>Dfd^{RNAi}* mutant conditions cannot be analyzed at 29°C.

As tension remains present in the neck in the absence of *Tollo* or Dg function and upon strong but partial reduction of Dfd, it is possible that additional downstream effectors of Dfd exist. We now state this in the manuscript: “While future work will be necessary to delineate how Dfd controls *Tollo* and Dg distributions and whether additional regulators act downstream of Dfd (Fig. 6h), the interplay between tissue tension and curvature likely expands beyond the fold we have explored here.” We have updated Fig. 6h* and its legends to reflect the possibility that other unknown downstream effectors of Dfd might promote neck invagination.

In addition, to help further understand the link between the upstream regulators and tension generation, the temporal profile of Dfd, *Tollo* and Dg expression in the neck region, in particular the earliest stage when they can be detected, should be described.

We now provide a description and corresponding images of Dfd, *Tollo* and Dg accumulation in the neck at 14, 18, 21 and 24 hAPF. For completeness and in line with the reviewer’s minor comment 2 (and with Reviewer 2), we now also provide the localization of apical and basal Ecad, MyoII and F-Actin at 14, 18, 21 and 24 hAPF. Since providing this dataset would occupy a large part of the figures and it would be interesting to view Ecad, MyoII, F-Actin, Dfd, *Tollo* and Dg together at different time points, we provide this dataset as a Supplementary Video 2*, in which all localizations can be viewed at 14, 18, 21 and 24 hAPF. We describe the dynamics of these markers over time in the figure legends.

Finally, it would be helpful to indicate the lateral border of Dfd expression in the neck domain and check whether it overlaps with the border of the high medial-lateral tension region of the neck domain.

The localization of Dfd and the tension on the most lateral region are difficult to access as imaging of this domain is hampered by the presence of the pupal head and eyes. Accordingly, we cannot measure tension on the most lateral region. To address this comment, we now report both Dfd accumulation and medial-lateral tension in the largest part of the neck domain accessible within the current technical limitations. This shows that the lateral tension is similar to the medial one (Extended Data Fig. 2d-f*) and that Dfd expression extends laterally (Extended Data Fig. 1b*).

2. The authors’ model argues for a critical role for actomyosin-dependent tissue tension in neck folding; however, disrupting myosin II or its activator Rok by RNAi only partially inhibited the folding process (Fig. 1k). While this could be explained by a partial knockdown effect, the lack of a stronger inhibition of actomyosin prevents the test of whether a complete loss of tension during the 16 – 24 hAPF window would block neck folding all together. The current data therefore could not entirely rule out the potential contribution of additional mechanism(s) that operates at later phase of the folding process. Is

there a way to completely abolish actomyosin contractility/in-plane tension prior to or during neck folding (e.g., treatment with Rok inhibitor or actin depolymerization drug)?

As suggested by the reviewer we have performed the Rok inhibitor (Y-27632) injection experiment and we have compared the dynamics of neck invagination in Rok inhibitor and mock (H₂O) injections. We found that invagination is completely stalled upon Y-27632 injection (Extended Data Fig. 2h*). Therefore, fully abrogating MyoII contractility is sufficient to prevent invagination. We state: "In addition, injection of the Rok inhibitor (Y-27623) during folding completely abolished invagination (Extended Data Fig. 2h)". This result agrees with the fact that in the *Dfd>Dfd^{RNAi}*, *Dfd>Tollo^{RNAi}* and *Dg* mutant conditions, the residual neck tension is sufficient to drive invagination.

Alternatively, a test of the immediate impact of disrupting tissue tension may be achieved by laser ablation at the medial region of the neck domain during the folding process, which should result in tissue recoil that substantially reduces tissue tension.

We understand that the reviewer expects that ablations at the medial region will reduce ML tension throughout the neck. We have performed such experiments and we observed that lateral tension is not reduced (for details see comment 4). Instead, laser ablation leads to a local effect and does not reduce stress throughout the neck. We note that this is not unexpected since in active materials where tissues and cells can generate mechanical stress autonomously, tissue stress could be maintained due to interaction with the extracellular matrix or other active processes. Based on these results and the above finding that Y-27632 fully abrogates invagination, we did not further explore the proposed experiments.

3. The way how Tollo and Dg/Dys activate actomyosin in the neck folding context is unclear. While delineating the entire pathway appears to be beyond the scope of this paper, the observation that disruption of Tollo (apical localized) or Dg/Dys (basally localized) affected both apical and basal tension to a similar extent is quite unexpected and worth additional investigation. As summarized in Extended Data Fig.4I, the authors propose the presence of feedback between apical and basal tensions. However, an alternative explanation is that Tollo and Dg/Dys are mutually dependent for their appropriate localization (and/or function). This possibility should be tested.

As suggested by the reviewer we have analysed the apical localization of Tollo in the absence of Dg function as well as the basal localization of Dg in absence of Tollo function. We found that apical localization of Tollo is independent of Dg and that basal Dg localization is not affected by Tollo loss of function. Therefore, the coupling between apical and basal tensions is unlikely caused by changes in Tollo or Dg localization. We have reported these results in Extended Data Fig. 5b,c* and indicate in the text: "Since Tollo is mainly localized at the apical side of the cells, independently of Dg activity, and Dg is enriched at the basal side of the cells even in the absence of Tollo activity (Extended Data Fig. 5b,c), this could suggest that apical and basal tensions feedback on each other, or that Dg and Tollo have unforeseen indirect roles on the apical and basal tension, respectively.". In line with this comment, we have also extended the discussion of the manuscript on the crosstalk between apical and basal tension as well as modified the original Extended Data Fig. 4I (now Fig. 6h*) to better indicate that the interplay between basal and apical tensions is not the only possible mechanism of coupling apical and basal tensions. We state in the discussion: "We found that apical Tollo and basal Dg regulate basal and apical tensions without controlling each other's localizations and we did not detect any short time scale (tens of seconds) mechanical coupling between apical and basal tensions (Extended Data Fig. 6i,j).

Therefore, we envision that the correlated apical and basal tensions are more likely linked to a long-time scale mechanical feedback or to an indirect role of Tollo or Dg on the basal or apical tension, respectively (Fig. 6h)”

4. The tissue ablation experiment presented in Figure 4i-4n is an important experiment designed to test the role of tissue curvature in neck folding. The phenotype is striking. However, at the current form, it is difficult to determine whether the observed effect on neck folding is due to direct alteration of curvature or due to some indirect effects. First, it is unclear at which stage the ablation was performed. If the neck region is already under medial-lateral tension at the time of ablation, removing the lateral sides of the neck region would be expected to cause recoil of the remaining neck tissue towards the midline, resulting in a reduction of medial-lateral tension. If this happens, it may complicate the interpretation of the results.

We thank the reviewer for raising these important points.

First, we now clarify in the main text that these ablations were performed prior to neck folding to ensure full disruption of the tissue prior to folding; and that they were repeated in time to prevent tissue repair. We state “Prior to neck folding, we used a UV laser to ablate the tissue across its whole thickness to mechanically isolate the medial region from the lateral curved regions (Extended Data Fig. 11f). We then imaged neck folding while repeating the tissue severing ablations every 3h (Fig. 6a-g and Supplementary Video 5).”

Second, to directly address this point we have performed the following experiment: Upon lateral severing of the tissue by UV laser ablation, we have probed the tension in the tissue when the tissue is flat and stops invaginating. We found that the ML tension is still high in the medial region of the tissue. This finding further strengthens our conclusion that curvature is necessary for invagination as the observed folding changes are unlikely to be due to an indirect effect on tissue tension. We note that this experiment emphasizes that the tension generation is local: while we perform the ablations prior to neck folding, the cut off tissue is still able to build up tension. The results are presented in Extended Data Fig. 11g,h* and we state in the manuscript: “When the isolated medial region was initially flattened by the coverslip, the tissue deepening speed was drastically reduced, while the tissue remained under ML tension (Fig. 6a-c and Extended Data Fig. 11g,h).”.

Second, considering potential secondary effect of laser ablation (e.g., generation of heat) on tissues neighboring to the ablated region, the authors should examine whether the treatment affects myosin/actin distribution and/or actomyosin-dependent tissue tension in the medial region of the neck.

As indicated above we have found that UV laser severing of the tissue on the lateral sides did not reduce tissue tension in the medial region of the tissue. This indicates that the impact of severing is local and does not prevent cells to generate mechanical stress.

Finally, evidence that the laser treatment physically isolated the medial neck region should be demonstrated.

To address this reviewer comments, we have first clarified in the manuscript that we used two types of ablations (See Methods, page 23, Lines 17): (i) The “probe ablation” where we use *an infrared laser in multi-photon mode* to locally ablate a thin layer of the tissue (i.e., either at the apical or basal cortex of the cells). This is used to measure recoil velocity as a proxy to tissue tension (Extended Data Fig. 2b*). (ii) “Tissue severing ablation” where we use *an ultraviolet laser in single photon mode* to illuminate the

tissue from apical to basal to sever the whole tissue along its apical-basal axis, and physically isolating the medial region from the more lateral ones (Fig. 6a,d*).

As indicated above, here we have performed "tissue severing ablation" to provide evidence that the latter laser treatment physically isolated the tissue region. We have performed this whole tissue thickness ablation in *Ecad::3xGFP* animals where the whole apical basal axis can be visualized. We now provide an image before and after tissue severing ablation showing the tissue is indeed fully cut throughout the apical-basal side, separating the medial tissue region from the lateral ones. This is also illustrated by the presence of very large autofluorescent yolk granules that are now localized at the most apical side of the tissue instead of being inside the pupal cavity (Extended Data Fig. 11f*).

5. Regarding the analysis designed to rule out the contribution of apical constriction/cell wedging (Extended Data Fig.5a – c), it is unclear why the authors select 20 hAPF and 22 hAPF for comparison. It would be more informative to compare the cell morphology between timepoints near the beginning and end of neck folding (e.g. 16 and 24 hAPF, respectively). The selection of stages could be important since in some other folding processes (e.g. *Drosophila* ventral furrow formation, ascidian endoderm invagination), cell wedging/basal expansion does not happen until later phase of folding. In addition, 3D cell shape should be shown to make the conclusion more convincing.

As requested by the reviewer we have quantified cell wedging between 16 hAPF and 24 hAPF along the AP orientation to further explore the role of apical constriction/basal relaxation perpendicular to the fold axis. Between 16 hAPF and 24 hAPF, we found that the apical domain keeps a similar AP width and that the basal domain reduces its width; thus, cell wedging during the invagination process is unlikely to be the main cause of folding (Extended Data Fig. 6c,d*). In addition, we now show the 3D cell shape in the neck region at 16 and 24 hAPF (Extended Data Fig. 6b*).

6. The physical model presented in this study considers a line under tension (Fig. 3c), however, the neck epithelium has a thickness that is non-negligible compared to the size of the neck fold. How tissue thickness may influence the comparison between the real tissue and the computer model should be discussed.

We agree that the physical model presented in this study considers a line under tension. We have chosen such simplification based on the following theoretical and experimental considerations. From the theoretical point of view, to know whether we can treat the neck as a line, we need to compare the neck thickness with the neck radius of curvature. Since the tissue thickness ($\sim 10\mu\text{m}$) is only 3% of the value of the neck radius of curvature ($\sim 300\mu\text{m}$), the tissue can reasonably be treated as a thin layer. From the experimental point of view, the physical model presented here is a data-driven analysis using the classical equation of Laplace force, of general validity, to perform predictions. Altogether, this model of a line under tension yields correct predictions of neck folding dynamics for control condition ($R^2=0.96$, Fig. 4d*) and the different mutant conditions plotted together ($R^2=0.78$, Fig. 4f*) in which mechanical tension is altered.

Therefore, while considering tissue thickness will greatly complicate the model, we expect that the improvement will be of a few percent relative to the already excellent agreement we have and will not *per se* help to explore the role of tissue curvature.

To improve our manuscript based on this reviewer comment, we first add in the main text: "Here, since the tissue thickness ($\sim 10\mu\text{m}$) is smaller than the neck radius of curvature ($\sim 300\mu\text{m}$), we applied the equation of Laplace force to a thin epithelial strip which moves inwards perpendicular to its surface (see

Supplementary Note).". In addition, we now fully discuss our line of argumentation regarding tissue thickness in the Supplementary Note (See Supplementary Note section 3b, page 29, line 40).

Minor points:

1. In Figure 1b, the authors show that the neck cells are already elongated medial-laterally at 18 hAPF, whereas the cells in the head/thorax regions are more isotropic. What causes this morphological difference between the neck and head/thorax regions? When does this difference first appear? Is *Dfd* (or *Tollo*, *Dg/Dys*) involved in generating the elongated morphology of the neck cells?

To address this point we have performed a detailed characterization of cell shape anisotropy between 13 hAPF and 22 hAPF in the neck, head and thorax; 13 hAPF being the earliest time point at which we can image the tissue with enough resolution (since we need to remove the pupal case to image *Ecad::3xGFP* at sufficient resolution). As shown on Extended Data Fig. 1a*, we found that neck cells are already elongated along the ML axis at 13 hAPF. We found that cell elongation increases at the time of neck invagination; this is consistent with the increase of ML tension (Extended Data Fig. 3c*). We have also looked at cell elongation in *Dfd>Dfd^{RNAi}*, *Dfd>Tollo^{RNAi}* and *Dfd>Dg^{RNAi}* mutant conditions. We found that the cell elongation is similar between control, *Dfd>Dfd^{RNAi}*, *Dfd>Tollo^{RNAi}* and *Dfd>Dg^{RNAi}* at 13 hAPF, but *Dfd>Dfd^{RNAi}*, *Dfd>Tollo^{RNAi}* and *Dfd>Dg^{RNAi}* cell elongation was less than control animals during invagination. As we have shown that these genes control ML tension, we favour the idea that the reduced ML cell elongation observed in these experimental conditions is caused by the reduced ML tension. We have added this dataset in Extended Data Fig. 3c*.

2. In figure 1g, h, only 18 hAPF is shown for apical signals, and 21 hAPF is shown for basal signals. To understand how apical and basal tissue tension evolve over time, it would be helpful to show how the apical and basal actomyosin structures change throughout the folding process.

As requested, we have now included the apical and basal distribution of *MyoII*, F-actin and *Ecad* at 14, 18, 21 and 24 hAPF. We observed that apical and basal actomyosin structures become more pronounced along the ML axis over time in agreement with the increase in ML tension and in cell elongation. We have added this data in as Supplementary Video 2* along with the *Dfd*, *Tollo* and *Dg* localizations.

3. The basal tissue tension is non-zero between 14 – 18 hAPF. This observation should be discussed in the light of the finding that "MyoII and F-actin fibers ... being strongly aligned with the ML axis from 21 hAPF onwards".

This basal tension likely arises from the basal actomyosin network that can already be observed at 14 and 18 hAPF as now shown in the Supplementary Video 2*. Accordingly, we now write in the caption: "MyoII and F-actin are observed to form a network at the basal side of the neck at 14 hAPF. The anisotropy of this actomyosin network increases over time, becoming noticeably pronounced by 21 hAPF in the medial region of the neck (c).".

4. Extended figure 2: the title "measurements of apical-basal tension" is misleading (it seems to indicate tension along the apical-basal axis). Better restated as "measurements of apical and basal tension".

We have corrected the Extended Data Fig. 2* title as indicated. **“Actin organization, measurements of apical and basal tensions and neck apical folding.”**

5. Extended figure 2i: The red line is missing from this panel. If the apical side moved out of the view at 24 hAPF, it should be stated in the figure legend.

We apologize for this omission, and we now indicate in the figure legends (Extended Data Fig. 2g*) that the apical side moves out of the field of view: **“Note that in the *Dfd>Mbs^{RNAi}* the tissue has moved out of the field view at 24 hAPF and the apical fold front line cannot be shown.”**

6. Extended figure 3b: The difference between the mutant clone (outside of the dotted circle) and the control region (within the dotted circle) in the *Dfd* RNAi sample is very hard to appreciate. If the intention is to compare between the *Dfd* RNAi sample and the control sample, it should be made clearer in the figure legend.

We agree that this Extended Data Fig. 3b was unclear since we showed the distribution of *Dg* in *Dfd^{RNAi}* clones in the neck region by comparison with control clones in the lower panel. As suggested, we have now updated these figure panels, which are now in main Fig. 3n*, and show *Dg::GFP* localization in *Dfd>Dfd^{RNAi}* and control conditions to better show that loss of *Dfd* function disrupts *Dg::GFP* localization in the neck region.

7. Extended figure 3b and 3c: The basal F-actin signal in the control in panel b appears very different from the basal F-actin signal in the *Dfd*-GAL4 control in panel c.

Addressing the above point 6 also corrected this point and the figure now shows that the F-actin organisation is not affected.

8. Extended figure 4k: what is the bright signal at the basal region of the cross-section view? (It is not present in panels 4f – 4j).

In the panel 4f-4j we have labelled the tissue using a GFP knock-in at the *Ecad* locus specifically expressing *Ecad::GFP* in epithelial tissue. In the panel 4k, for genetic technical reasons we had to use a *Ubi-Ecad::GFP* transgene that leads to GFP signal expression in all cells of the animal including the circulating fat cells or hemocytes under the epithelium. The bright GFP signal are circulating cells labelled by *Ecad::GFP* signal. To clarify the origin of this difference, we now state in the figure legend (Extended Data Fig. 4d*): **“Tissues are labelled with *Ecad::3xGFP*, except for *Dg* which is labelled with *ubi-Ecad::GFP*. The *ubi:Ecad::GFP* transgene also promotes *Ecad::GFP* expression in circulating cells that are present below the tissue.”**

9. Extended figure 5f: It is unclear whether ablation occurred or the tissue was simply photobleached. The conclusion would be more convincing if the authors could show evidence that the lateral cell cortex was indeed ablated.

In these experiments we used a laser power that is identical to the one used for basal ablation, which is sufficient to robustly ablate the actomyosin network and trigger tissue recoil in a much deeper location. We can therefore safely state that we are not just photobleaching the tissue. To clarify this

point we now state in the figure legends (Extended Data Fig. 6g*) and the Methods: "Since this laser power is sufficient to ablate the actomyosin network of the tissue at a deeper position to trigger tissue recoil, we can safely consider that this regime is sufficient to ablate the lateral cortex to estimate recoil velocity. "

10. Extended Figure 6: The purpose of this figure is to show the myosin distribution; however, only images with Ecad-GFP signal were shown. Myosin images should also be displayed in addition to the quantification plot.

We would like to clarify that the purpose of this figure was to illustrate the position of the ablations performed in the lateral and medial regions of the neck. Following the reviewer comments, we have reorganized the figure panels: the positions of the medial and lateral ablations are shown in Extended Data Fig. 2d* and the distributions of Ecad and MyoII are shown in the Supplementary Video 2*.

11. Extended Figure 9a: it is interesting that Rok RNAi and Myosin II RNAi mainly influence the basal tension after 18 hAPF, but have no effect on the initial tension increase between 14 – 18 hrs.

As Rok and MyoII are well known to control tissue tension, this might reflect an incomplete knock-down at earlier time points of the experiments. Indeed, these experiments are performed by inducing the dsRNA expression by a temperature shift from 25 to 29°C and the downregulation of Rok and MyoII might not be as complete at these earlier time points. We now indicate in the Methods (page 21, line 7) that: "As in any temperature shift experiment using the Gal4 system, the magnitude of the loss-of-function depends on the duration of RNAi induction. The observed phenotypes might therefore be stronger at later time point of development."

Reviewer #2:

The authors described for the first time the formation of neck folding between head and thorax in *Drosophila* pupae and revealed a new mechanism of fold formation including contribution of acto-myosin-dependent in plane tension (both in basal and apical sides) and medio- lateral (M-L) curvature of the tissue. They combine live imaging and laser ablation in different genetic backgrounds to describe the contribution of in plane tension and curvature of the head-thorax neck fold formation during pupal development. The paper is organized in two parts: First, they identify the homeotic gene *Dfd* as a key regulator of both apical and basal in plane tension through respectively *tollo* receptor expression and *Dg* and *Dys* localization in the presumptive neck compartment. Then, they found a correlation between curvature along M-L axis and deepening speed and addressed the contribution of curvature in tissue depth invagination and deepening speed.

Overall, I found that the results described here are very interesting and of high quality. They should certainly be published in Nature Communication. However, I have a number of concerns that I think the authors must address before publication.

We thank the reviewer for her/his positive comments regarding the significance and quality of our work as well as her/his detailed and constructive review that helped us to improve our manuscript. We have addressed point by point her/his comments below.

Experimental part²:

1. Fig1 Major point: Splitting of current Fig1 in two figures would allow for a more complete description of the tissue's properties.

The first as a complete description of the *drosophila* neck folding during pupal development: since this invagination has never been described, it is necessary to show cell shape, *Dfd* expression and Myosin accumulation at the initial point just prior fold formation (14H APF global view and zoom as in Fig1 2b). If *Dfd* regulates apical and basal tension in these cells, it would be interesting to visualize *Dfd* expression and DE-Cad expression (cell shape) from the onset of the process (14H) and up to 24H.

As requested, we have split Figure 1 in two figures. Also, we now provide a more complete description of the neck folding during pupal development by reporting the *Dfd* and *Ecad* distributions as well as apical and basal localizations of MyoII, and F-actin at 14, 18, 21 and 24 hAPF. In addition, we now include *Tollo* and *Dg* localization at 14, 18, 21 and 24 hAPF. Since providing this dataset would occupy a large part of the figures and it would be interesting to view *Ecad*, MyoII, F-Actin, *Dfd*, *Tollo* and *Dg* together at different time points (as requested below and, also by the Reviewer 1), we provide this dataset as a Supplementary Video 2^{*3}, in which all localizations can be viewed at 14, 18, 21 and 24 hAPF.

It would be also interesting to describe the mechanical properties of the *Dfd* expressing compartment in comparison with head and thorax tissues (in sup data).

² We have numbered this reviewer's comments to ease cross refereeing between them.

³ Fig. or Extended Data numbers with an asterisk refer to the revised manuscript. The text modifications addressing the reviewers' comments are in green in the revised version of the manuscript.

We now provide a comparison of the mechanical properties in the head, thorax and neck tissues at 22 hAPF. To this end we have measured recoil velocities upon ablation along the AP and ML axis in the head, thorax and neck. This shows that the medial-lateral tension is higher in the neck than in the head and thorax tissues, whereas AP tensions are similar. These findings agree with a role of ML tension in driving the invagination process. These data are shown in Extended Fig. 2a* and we state in the main text: **"The flanking head and thorax tissues also exhibited very little AP tension, and notably much lower ML tension compared to the neck region (Extended Data Fig. 2a)."**

Fig1d might be in sup data.

Since we have some room in Fig. 1 and we believe that it supports the main line of argumentation, we keep Fig. 1d as a main figure.

The title of the second figure could be: Drosophila neck folding required Myosin- dependent apical and basal in plane tension. Apical and basal MyoII accumulation should be described for 3 time points (14H, 18H and 21H for example). As mentioned above, showing the initial state (14H) would be informative.

We agree that this figure title needed improvement and we thank the review for her/his suggestion. The proposed title could be interpreted as if we had independently modulated apical and basal MyoII contractility. We therefore choose a new title: **"Dfd accumulation and MyoII-dependent tension during neck folding."** As indicated above we now provide this apical and basal MyoII dataset in Supplementary Video 2*.

2. Major points: Concerning ablations reported in Fig1 I and j, it is not clear in which part of the tissue ablation is done (lateral, medial or pool of different locations?).

We apologize the initial Fig. 1i,j legends were indeed imprecise regarding the position of the ablation. In the revised legend (Fig. 2c,d*) we now write: **"(c,d) Graph of the ML apical (c) and basal (d) initial recoil velocities (mean \pm sd, averaged with a 2h sliding window) upon ablation in the medial neck region as a function of developmental time (N=127 pupae). See also Supplementary Video 3."**

Apical and basal tension must be presented in central versus medial in the neck regions, as currently done in extended data fig6f and g.

We agree that extending this dataset is relevant. We have thus performed apical and basal ablations in the lateral regions over the time course of development and plotted them in Extended Data Fig. 2e,f* along with the medial recoil velocity measurements. This shows that lateral and medial tensions similarly increase during development. We now indicate in the text: **"In addition, the apical and basal ML recoil velocities measured in the medial versus lateral neck regions were similar (Extended Data Fig. 2d-f)."**

3. What about contribution of the head-thorax apical myosin cable? Is it cut in the apical ablation experiments? There is a difference in basal and apical tension, especially at 14H. Could it be due the ablation (or non-ablation) of the apical myosin cable? This difference in tension between apical and basal should be discussed. (discussion).

From the perspective of our model the roles of the cable and the neck domain are similar, and therefore we did not aim at distinguishing here their relative contributions. Accordingly, in recoil velocity measurements in wt and mutant conditions to measure tissue ML tensions, we have cut both the actomyosin cable at the interface between the thorax and the neck as well as the cells in the neck domain. Therefore, the difference in apical versus basal tension is unlikely to be caused by the non-ablation of the cable. As we have estimated tissue tension using recoil velocity measurements, the difference in apical versus basal tension can be due to the difference in mechanical properties (such as viscosity or friction on the ECM) of the apical versus basal tissue. We agree that it would be interesting to explore the respective contributions of the cable and the apical domain in the future. As suggested by the reviewer we now discuss this point in the discussion of the manuscript: "An additional novel feature unveiled by our work relates to the presence of two types of supracellular actomyosin structures in the apical region: the apical actomyosin cable at the boundary between neck and thorax and the strong actomyosin enrichment in the neck cells. We did not aim at separating the contributions of these two supracellular structures, and instead targeted to decipher the role of tissue curvature. Indeed, our theoretical model used to explore the role of tissue geometry gives very good predictions when treating the neck as a line and measuring recoil velocities across the two structures. We foresee that the characterization of the putative respective roles of these two actomyosin structures could help to delineate how (i) the boundary between homeotic compartments is controlled, (ii) the detailed sagittal shape of the invagination is modulated, and (iii) the feedback between apical and basal tensions is controlled."

4. Minor point: In extended fig1 I wonder how do you define the neck region in panel a. Head cells next to the neck region appears elongated along the M-L axis. Also, it should be informative to delimit in the sup Fig1c panel the neck region, so readers have a notion of the size of the neck relative to the head.

We agree that the red overlay was misleading. It did not highlight the neck region, but rather the region at the neck-thorax interface corresponding to a 20 μ m box used to track the apical fold front in transverse sections. To improve this figure (now Extended data Fig. 1c*), we have removed the red overlay and indicated this region by a red bracket on each side of the image. Thereby, the elongated nature of the neck cells as compared to the thorax cells can be better appreciated. We choose to track the neck position on a transverse section at the interface between the thorax and neck cells as it can be unambiguously defined by the changes in cell shapes, while the head-neck interface can be more difficult to delimit without the Dfd localization. As suggested by the reviewer we now indicate the position of the neck by a magenta bracket as in other figure panels.

5. Fig2: Apical Tollo accumulation is clear and down regulation of Myo in RNAi Tollo clone as well. However, Dg:GFP basal accumulation within neck region is not clear (Fig 2e and extended data fig4b). Dg must be presented in a more global view, together with Dfd expression, to better visualize Dg fibers within Dfd expressing cells.

We have improved this dataset by showing a more global view of Dg with a Dfd staining. This is shown in Supplementary Video 2*. This clearly shows that Dg is present in the neck and extends in the head and thorax tissues.

6. If possible, it would be better to downregulate Dg or Dys within the Dfd expressing cell (using RNAi driven by DfdGal4) and not in the whole tissue since their expression is not specific of the neck region (extended data fig 4b and c).

We thank the reviewer for suggesting this experiment. We have downregulated Dg function within the Dfd domain (using *Dfd>Dg^{RNAi}*). We observed that invagination speed is reduced relative to the *Dfd-GAL4* control. This further confirms the role of Dg within the neck domain. We have added this result in Extended Data Fig. 4f* and we now state in the main text: "...we found that loss of Dg or Dys function (using *Dg* and *Dys* null mutant animals) as well as inhibiting Dg function in the neck (*Dfd>Dg^{RNAi}*) also slowed down folding (Fig. 3h and Extended Data Fig. 4d,f)".

7. In Fig 2j actin intensity must be commented, in addition to actin fiber orientation.

Checking the Fig 2j raw data we realized that the gains on the Fig 2j had been poorly adjusted for display when preparing the figure; thus conveying the impression that phalloidin staining was weaker in *Dg* mutant conditions. To address this comment, we have therefore corrected the gain of Fig 2j (now Fig. 3k*) and commented that phalloidin intensities are similar in control and *Dg* conditions. We state in the figure legend: "While F-Actin fiber orientation is affected in *Dg* mutant tissues, the phalloidin signal level is similar in control and *Dg* mutant tissues."

8. When Dg or Dys are downregulated, very few tension is observed in basal as in apical but invagination still occurs (fig 2 g,h i). This result must be commented. May be this could be due to the tissue flow shown in extended data fig5. (discussion)

We note that a change in recoil velocity could be interpreted as a change in the tension or in mechanical properties such as the friction or viscosity. Therefore, we believe that it is difficult to fully interpret the difference between Dg and other mutant conditions. This being said, we fully agree that it will be very interesting to explore the permissive versus active role of the surrounding tissue. As suggested by the reviewer on this point and the one below, we now comment on this aspect in the main text: "We note that the slope of the linear relationship between the deepening speed and the product of estimated tissue ML tension and curvature varies between wild-type (wt) and some mutant conditions (Extended Data Fig. 8c). Since we assessed tissue tension via recoil velocity that depends upon tension as well as tissue mechanical properties (friction and viscosity)^{33,42}, the slope variabilities may indicate that some mutant conditions could also affect tissue mechanical properties. This is for example the case for Dg loss of function exhibiting a higher slope than the one obtained in wt conditions (Extended Data Fig. 8c), as it promotes a strong decrease of recoil velocity while having a more modest impact on invagination dynamics (Fig. 2h-j)". In addition, and in response to the comment 17 below, we have extended the discussion regarding the role of tissue flows (see below).

9. Extended data Fig3: minor point Results in a is clear but it is not the case in b. Panel b must be deleted. Actin accumulation is already shown in c (*Dfd>DfdRNAi*). It would be necessary to show Dg:GFP accumulation depending upon Dfd, but it could be easier to show it in *DfdGal4> uasDfdRNAi*.

We agree that the figure layout was unclear. The upper b panel was showing Dg::GFP localization in *Dfd^{RNAi}* clones that almost cover the full neck region. This was illustrating the loss of Dg accumulation in the absence of Dfd function by comparison with control clones shown below. As the Dfd and control clones were covering most of the neck tissue, we agree that showing Dg localization in *Dfd>Dfd^{RNAi}* is simpler, and we have therefore changed the dataset as suggested by the reviewer. These new data are now reported in Fig. 3n*.

10. Extended data Fig4: Minor point: The model in L must be presented in Fig2. The data presented

in apical show clearly that Dfd might regulate Tollo probably directly but note that in basal Dfd might regulate Dg and Dys localization.

As suggested, we now present the model as a main figure (Fig. 6h*). In line with the request to expand the discussion (comment 17 below), we have also updated this figure and included a more general model summarizing the contributions of genetic patterning and tissue curvature in regulating fold invagination. We agree that Dfd might differently regulate Tollo and Dg/Dys function. We now state in the figure legends: "Dfd might regulate Tollo and Dg accumulation via either similar or distinct mechanisms...".

11. Extended fig5: Major point: In j, ablation at 17H APF reduces depth invagination indicating that flow might contribute to initiation of the process (reduction of depth is comparable to the one observed with Dg or Dys mutants Fig 2g). Please discuss this point in the discussion.

As mentioned above we fully agree that the permissive or active contribution of tissue flow would be very interesting to explore in the future. As suggested, we now comment on this point in the discussion by stating: "Finally, a last feature relates to the large scale of the neck fold we studied, and thus, the permissive versus active contributions of the neighboring tissue flows. Since the neck fold is very large relative to the size of the cells, the head and thorax tissue flows are substantial relative to previous folding processes driven by apical constriction^{7,8}. Our ablation of head and thorax tissues illustrated that on the timescale of 5-6 hours (Extended Data Fig. 6n,o and Supplementary Note), neck folding is not driven by tissue buckling due to the opposite cell flows of these two tissues, agreeing with a permissive role of the head and thorax flows. On longer time scale, around 6h post ablation, the deepening speed started to decrease relative to non-ablated control tissue, presumably as a long-term consequence of the tissue stretching observed in the head and thorax cells near the ablation sites (Extended Data Fig. 6n). In the future it will be interesting to explore whether over long time scales surrounding tissue flows contribute to large fold formation during development."

12. Fig4: Major points: In d the recoil velocity values used from Fig1i are measured in a flattened tissue, although the tissue is not flattened this time. Please use recoil velocity measured in the non-flattened tissue to present your graph. These measures could be presented in extended data.

We thank the reviewer for raising this important point. We had initially used the measurements of tension in flattened animals since the measurements of non-flattened animals were challenging. To address this comment, we have improved the mounting of non-flattened animals and we are now able to measure tension in this context. We have then generated a new set of experiments to measure tension in non-flattened animals. We now show this complete data set in Extended Data Fig. 11a,b*. This shows that the tension in non-flattened animals is similar, albeit a bit lower than the one in flattened animals. Interestingly, despite the slightly lower tension, the non-flattened tissue invaginates faster, further arguing for the key role of tissue curvature in controlling the dynamics of neck invagination.

13. The curvature must be presented in non-flattened tissue in c.

We have added the requested graph in Fig. 5d*.

14. In h, I wonder what is the best recoil velocity data set to use here from flattened (currently presented in Fig 1i) or non-flattened tissue (see above)?

Building on this reviewer comments and on the previous one, we decided to set a curvature threshold to select whether the flattened or non-flattened recoiled velocities are used to compute the product of curvature and recoil velocity in Fig. 5j*. We have set this threshold to $0.00015\mu\text{m}^{-1}$ based on the difference in curvature between the medial and lateral regions upon coverslip flattening (see Fig. 5i*). We found that these refinements in our analysis do not change our initial conclusion regarding the strong correlation between curvature and the product of recoil velocity with the invagination speed ($R^2=0.83$ with curvature threshold versus $R^2=0.81$ with tension only measured in flattened conditions). In addition, we have checked that the curvature threshold can be chosen between $0.0005\mu\text{m}^{-1}$ and $0.002\mu\text{m}^{-1}$ without significantly affecting the conclusion since R^2 varies between 0.81 to 0.83. We have updated the Fig. 5j* according to this reviewer's suggestions and we now indicate the threshold used in the figure legends: "Recoil velocities measured in flattened animals (Fig. 2c) were used for regions of initial curvatures below a threshold of $0.00015\mu\text{m}^{-1}$, while recoil velocities measured in non-flattened animals (Extended Data Fig. 11a) were used for regions of initial curvatures larger than $0.00015\mu\text{m}^{-1}$. A line passing through the origin was fitted ($R^2=0.83$). Using a curvature threshold equal to 0 or varying it from $0.0005\mu\text{m}^{-1}$ to $0.002\mu\text{m}^{-1}$ leads to R^2 varying from 0.81 to 0.83."

15. Minor point: The result with lateral flattening must be more exploited in terms of central versus lateral tension. In i comment the fact that in the neck region cells are still elongated, suggesting that they are still under ML tension, whereas invagination does not occur.

We understand that the referee is referring to two different figures, 4e and 4i.

Regarding Fig. 4e (Fig. 5f* in the revised manuscript), which shows 'lateral flattening': In the original and the revised manuscript we describe the implication of the lateral flattening: "If curvature modulates folding dynamics, reducing curvature in any region of the fold should locally reduce deepening speed. We thus flattened the lateral side of the neck region by mounting the pupa on its side; and used the contralateral non-flattened side as an internal control (Fig. 5f and Supplementary Video 4). In these experimental conditions, the flattened lateral side behaves like animals imaged with coverslip flattening in the medial domain, while the control non-flattened lateral side deepens with faster dynamics (Fig. 5g-i and Extended Data Fig. 11c). Altogether, we conclude that modulating the curvature led to changes in deepening speed that are predicted by the product of the curvature and the tension (Fig. 5e,j)". In addition, in response to the comment 1 we now show that lateral and medial tensions are similar (Extended Data Fig. 2d-f*). We therefore believe that we have fully interpreted these important findings. If the reviewer believes that the text needs additional modifications, then we will be happy to do so.

Regarding Fig 4i (Fig. 6a* in revised manuscript), which shows 'lateral ablations': We thank the reviewer for making this observation and constructive comment. To directly confirm that the cells were under tension, we have now performed experiments to measure tissue recoil velocity after the lateral severing of the tissue. We found that the tissue tension remains high, which is compatible with the role of tissue curvature to drive invagination. The results are presented in Extended Data Fig. 11g,h*.

16. Extended data fig10: minor point: d panel might be transferred in figure 4.

As suggested, we have transferred the Extended Data Fig 10d panel to Fig. 6g* (former figure 4).

17. Major point: Discussion should be developed, including some of the points mentioned previously. Interplay between apical and basal tension is very interesting. The authors could propose an hypothesis to account for the interplay between apical and basal tension.

We have developed the discussion along the lines proposed by the reviewer. In particular, we now better discuss several hypotheses that could account for the interplay between apical and basal tension based on the following experimental evidence. We have now added data to show that Tollo function does not impact Dg accumulation and vice-versa that Dg does not control Tollo localization. Our sequential ablation of the apical and then basal tension and vice-versa demonstrate that apical and basal tensions are not mechanically coupled on short timescale (Extended Data Fig. 6i,j*). We have therefore added the following statements in the discussion. "We found that apical Tollo and basal Dg regulate basal and apical tensions without controlling each other's localizations and we did not detect any short time scale (tens of seconds) mechanical coupling between apical and basal tensions (Extended Data Fig. 6i,j). Therefore, we envision that the correlated apical and basal tensions are more likely linked to a long-time scale mechanical feedback or to an indirect role of Tollo or Dg on the basal or apical tension, respectively (Fig. 6h), via for example the apical-basal microtubule networks. The characterization of the putative mechanical feedback and the potential indirect roles of Tollo and Dg will be important to better understand how tissue folding and tissue thickness are controlled during neck morphogenesis, and more generally how the crosstalk between apical and basal tension is regulated in epithelial tissues.". To be consistent with the additional information and the longer discussion we have also further detailed the model figure (Fig. 6h*).

18. The flattening conditions might induce compression on the tissue. Therefore, all the experiments that modulate curvature could also alter concomitantly at least another property of the tissue. It must be discussed at the end of the ms.

We thank the reviewer for raising this point. In the initial version we had only discussed the possible impact of the flattening on the friction and adhesion: "... this proportionality relationship implies that the local modulation of neck folding dynamics with coverslip flattening is unlikely to be explained by the friction or the adhesion between the apical ECM and the epithelium in the regions flattened by the coverslip (see Supplementary Note).".

We agree that the coverslip might induce tissue compression and we have discussed the possible impact of tissue compression in details with other possible impacts of tissue flattening in the Supplementary Note (section 3d). In particular, we added: "The flattening conditions might induce compression on the tissue. Therefore, in principle all the experiments that modulate curvature could also alter concomitantly at least another property of the tissue. In particular, the flattening we impose slightly affects the tension (Extended Data Fig. 11a,b). In principle, it could also affect the mechanical properties of the neck and surrounding tissues, especially through mechanotransduction. Here, the relative change of surface due to the flattening, estimated by comparing the initial surface length before flattening (the arc of circle) with the flattened surface width (the corresponding chord) is of order of $3.7 \pm 0.5\%$. The shear deformation, estimated by the ratio of tissue surface displacement towards the pupa center to the flattened surface width, is of order of $10.6 \pm 1.3\%$. This is well within linear elasticity⁷⁹, "linear" meaning here that the effects are proportional to the cause, and hence likely to remain of order of a few percent too: the modification of mechanical properties is expected to remain moderate. We note that tissue thickness at the midline measured at 18 hAPF has a large variability and is $8.6 \pm 1.8\mu\text{m}$ for flattened animals ($N=10$) vs $7.5 \pm 1\mu\text{m}$ for non-flattened ones ($N=10$); hence the effect of flattening, which is to increase rather to decrease the tissue thickness, is not significant ($p=0.11$). In addition, what we ultimately check is that even if the values of tension, curvature or velocity change, their correlation remains and is compatible with the equation of Laplace force, and thus, with a role of curvature in tuning the spatiotemporal dynamics of neck invagination.".

We choose to place this information in the supplementary note to have a complete and self-contained section regarding the possible mechanical impacts of flattening.

In addition, we now state in the main text: "...this proportionality relationship implies that the local modulation of neck folding dynamics with coverslip flattening is unlikely to be explained by changes in the friction or the adhesion between the apical ECM and the epithelium in the regions flattened by the coverslip (see Supplementary Note). Other mechanical changes indirectly induced by the flattening are equally unlikely due to the small magnitude of the resulting compression (see Supplementary Note).".

19. Minor point: Finally, the neck region is frequently called homeotic compartment, as in the title of the ms or in the last chapter. However, homeotic regulation is not the main point of the ms, so authors could remove mention to this when not clearly tested (as is the case in the last chapter).

We have corrected the title of the last chapter to "*Tissue curvature regulates the spatiotemporal dynamics of folding.*".

REVIEWERS' COMMENTS

Reviewer #2 (Remarks to the Author):

This is a very thorough revision. The authors have carefully addressed all my previous questions. I fully support the publication of this elegant piece of work in Nature Communications.

Below are some suggestions to the authors regarding the new data:

1. Supplementary Video 2: It is very challenging to examine the localization data for multiple proteins in a movie. I had to open the movie as an image sequence and look at it frame by frame in order to see the details. I understand the reason why the authors chose to present this set of data in the current format, but I think presenting them as figures will work much better. If putting all panels into a single figure takes too much space, the authors may consider presenting panel a – c for all stages in one supplementary figure, and panel d - e in another supplementary figure.
2. Extended Data Fig 3a: the following description in the figure legend is a little misleading: First, "Dfd signal can be detected in the neck in Dfd>DfdRNAi mutant tissues and the neck region appears enlarged in this mutant condition." The authors might want to emphasize that ONLY A LOW LEVEL OF Dfd signal can still be detected in the neck in Dfd>DfdRNAi mutant tissues. Second, "In particular, the most anterior domain of the Dfd expression domain does not seem to be affected." It is unclear what is not affected. Finally, it is unclear whether the apical or basal actin is shown in the figure.
3. Extended Data Fig 3c: the meaning of the error bars should be indicated (s.d. or s.e.m.).
4. Extended Data Fig 5b: the impact of Tollo RNAi on basal actin organization is not very easy to appreciate in the figure. It would be helpful to use arrows to highlight the difference between the control and Tollo RNAi samples.
5. Based on Supplementary Video 2, Dfd, Tollo and Dg are already present in the neck at 14 hAPF. However, the apical tension only becomes detectable between 14 and 15 hAPF (Extended Data Fig 2e), and the increase in basal tension only starts around 17 hAPF (Extended Data Fig 3f). It would be an interesting point for discussion how the timing of tension increase is determined. In addition, apical Myosin signal intensity appears to peak at 18 hAPF, yet the apical tension continues to increase from 18 hAPF to 22 hAPF (Extended Data Fig 2e). A brief discussion about this apparent discrepancy could be helpful.
6. In Extended Data Fig 3c, the authors showed that the elongation of the neck cells along the ML-axis, which is measured as apical aspect ratio (ML/AP), increased from ~ 3 to ~ 6 from 16 hAPF to 20 hAPF. However, in Extended Data Fig 6c and 6d, the authors showed that the apical cell length along the AP axis and apical cell length along the ML axis are not significantly changed from 16 hAPF to 24 hAPF. These observations seem to indicate that the apical domain of the neck cells is less elongated along the ML-axis at 24 hAPF compared to 20 hAPF. Is this the case?

Reviewer #3 (Remarks to the Author):

The authors clarified/ modified all the points accordingly with our suggestions. They performed all the requested experiments and completed the discussion as proposed. I found the quality and quantity of data impressive and well-discussed. In conclusion, the paper must be accepted for publication in Nature Communication.

Reviewer #2:

This is a very thorough revision. The authors have carefully addressed all my previous questions. I fully support the publication of this elegant piece of work in Nature Communications.

We thank the reviewer for his/her positive comments and full support for publication.

Below are some suggestions to the authors regarding the new data:

1. Supplementary Video 2: It is very challenging to examine the localization data for multiple proteins in a movie. I had to open the movie as an image sequence and look at it frame by frame in order to see the details. I understand the reason why the authors chose to present this set of data in the current format, but I think presenting them as figures will work much better. If putting all panels into a single figure takes too much space, the authors may consider presenting panel a – c for all stages in one supplementary figure, and panel d - e in another supplementary figure.

We have extensively tried to present the data as proposed by the reviewer. Yet we cannot maintain the linearity of figure calling. Moreover, we believe it is important to be able to compare the different localizations at high resolution. The reader loses this possibility by the reviewer's proposition. Therefore, we maintain the data as a movie. To take into account this reviewer comment, we indicate in the Movie legend that, if preferred, the Movie can also be opened as a TIFF file in Fiji by stating: "Note that Movie 2 can also be opened as a stack in Fiji for easier visualization."

2. Extended Data Fig 3a: the following description in the figure legend is a little misleading: First, "Dfd signal can be detected in the neck in Dfd>DfdRNAi mutant tissues and the neck region appears enlarged in this mutant condition." The authors might want to emphasize that ONLY A LOW LEVEL OF Dfd signal can still be detected in the neck in Dfd>DfdRNAi mutant tissues. Second, "In particular, the most anterior domain of the Dfd expression domain does not seem to be affected." It is unclear what is not affected. Finally, it is unclear whether the apical or basal actin is shown in the figure.

We have indicated in the figure the most anterior domain of Dfd expression by blue brackets and rewritten part of the figure legends associated with Supplementary Fig. 3a to better clarify the effect of Dfd>Dfd^{RNAi} on Dfd localization. We now state; "At high gain, some Dfd signal can be detected in the neck in Dfd>Dfd^{RNAi} mutant tissues and the neck region appears enlarged in this mutant condition. In particular, the most anterior domain (blue brackets) of the Dfd expression domain does not seem to be affected." Furthermore, we added in the legends that basal F-actin localization is shown.

3. Extended Data Fig 3c: the meaning of the error bars should be indicated (s.d. or s.e.m.).

We have corrected the omission.

4. Extended Data Fig 5b: the impact of Tollo RNAi on basal actin organization is not very easy to appreciate in the figure. It would be helpful to use arrows to highlight the difference between the control and Tollo RNAi samples.

In the manuscript we do not claim an effect of Tollo^{RNAi} on the basal actin organization. Therefore, we cannot highlight any differences.

5. Based on Supplementary Video 2, Dfd, Tollo and Dg are already present in the neck at 14 hAFP. However, the apical tension only becomes detectable between 14 and 15 hAFP (Extended Data Fig 2e), and the increase in basal tension only starts around 17 hAFP (Extended Data Fig 3f). It would be

an interesting point for discussion how the timing of tension increase is determined. In addition, apical Myosin signal intensity appears to peak at 18 hAPF, yet the apical tension continues to increase from 18 hAPF to 22 hAPF (Extended Data Fig 2e). A brief discussion about this apparent discrepancy could be helpful.

We have previously stated that we cannot fully interpret differences in apical and basal tensions; "We note that we have used laser ablation to probe mechanical tension as validated in multiple experimental contexts^{33,42}. Therefore, it remains difficult to fully explain small differences in apical and basal recoil velocities as they could reflect differences in the apical versus basal tissue mechanical properties, due to distinct friction on the apical versus basal ECM, for example.". Hence, we do not believe that there is any apparent discrepancy. We have now improved this statement by indicating:" Therefore, it remains difficult to fully explain all temporal variations and differences in apical and basal recoil velocities as they could reflect modifications in the apical versus basal tissue mechanical properties, due to distinct friction on the apical versus basal ECM, for example.". We also state: "Besides, future works will be necessary to apprehend how apical and basal tensions are temporally controlled."

6. In Extended Data Fig 3c, the authors showed that the elongation of the neck cells along the ML-axis, which is measured as apical aspect ratio (ML/AP), increased from ~ 3 to ~ 6 from 16 hAPF to 20 hAPF. However, in Extended Data Fig 6c and 6d, the authors showed that the apical cell length along the AP axis and apical cell length along the ML axis are not significantly changed from 16 hAPF to 24 hAPF. These observations seem to indicate that the apical domain of the neck cells is less elongated along the ML-axis at 24 hAPF compared to 20 hAPF. Is this the case?

We have measured the average cell elongation (apical cell length) at 20 and 24 hAPF, which are $11.82 \mu\text{m} \pm 0.29 \text{ sem}$ and $11.64 \mu\text{m} \pm 0.42 \text{ sem}$, respectively. The reviewer is fully correct in her/his observation. This can also be appreciated by looking at the Supplementary Movie 2, where one can see that the cells do not visually increase their ML length between 21 and 24 hAPF.

Reviewer #3 (Remarks to the Author):

The authors clarified/ modified all the points accordingly with our suggestions. They performed all the requested experiments and completed the discussion as proposed. I found the quality and quantity of data impressive and well-discussed. In conclusion, the paper must be accepted for publication in Nature Communication.

We are grateful to the reviewer for his/her very strong support for publication.